# Characterised Flavin-Dependent Two-Component Monooxygenases from the CAM Plasmid of *Pseudomonas putida* ATCC 17453 (NCIMB 10007): ketolactonases by Another Name

**DOI:** 10.3390/microorganisms7010001

**Published:** 2018-12-20

**Authors:** Andrew Willetts

**Affiliations:** 1College of Life and Environmental Sciences, University of Exeter, Exeter EX4 4QG, UK; a.willetts@ex.ac.uk or andrewj.willetts@btconnect.com; Tel.: +44-7966 9684-87; Fax: +44-1392-8513-04; 2Curnow Consultancies, Helston TR13 9PQ, UK

**Keywords:** ketolactonase, diketocamphane monooxygenase, flavin-dependent two-component monooxygenase, enantiocomplementary enzymes, enantiodivergent biotransformation

## Abstract

The CAM plasmid-coded isoenzymic diketocamphane monooxygenases induced in *Pseudomonas putida* ATCC 17453 (NCIMB 10007) by growth of the bacterium on the bicyclic monoterpene (*rac*)-camphor are notable both for their interesting history, and their strategic importance in chemoenzymatic syntheses. Originally named ‘ketolactonase—an enzyme system for cyclic lactonization’ because of its characterised mode of action, (+)-camphor-induced 2,5-diketocamphane 1,2-monooxygenase was the first example of a Baeyer-Villiger monooxygenase activity to be confirmed in vitro. Both this enzyme and the enantiocomplementary (−)-camphor-induced 3,6-diketocamphane 1,6-monooxygenase were mistakenly classified and studied as coenzyme-containing flavoproteins for nearly 40 years before being correctly recognised and reinvestigated as FMN-dependent two-component monooxygenases. As has subsequently become evident, both the nature and number of flavin reductases able to supply the requisite reduced flavin co-substrate for the monooxygenases changes progressively throughout the different phases of camphor-dependent growth. Highly purified preparations of the enantiocomplementary monooxygenases have been exploited successfully for undertaking both nucleophilic and electrophilic biooxidations generating various enantiopure lactones and sulfoxides of value as chiral synthons and auxiliaries, respectively. In this review the chequered history, current functional understanding, and scope and value as biocatalysts of the diketocamphane monooxygenases are discussed.

## 1. Introduction

The significance of the large transmissible CAM plasmid present in *Pseudomonas putida* ATCC 17453 (P, PpG1, C1, NCIMB 10007) in coding for a number of the enzymes necessary for the catabolism of the bicyclic monoterpene camphor was first recognised from the pioneering studies undertaken by Irwin Gunsalus’ research group at the University of Illinois, commencing in the late 1960s [1,2]. The discovery evolved out of a conceptually broader programme of research initiated to study carbon cycling in the biosphere which focussed specifically on a study of the biochemical challenges posed by the degradation of alicyclic camphor to short-chain aliphatic intermediates suitable to gain entry into the central pathways of metabolism via the tricarboxylic acid (TCA) cycle. This body of research (review: [3]) had already established that several different monooxygenases (Figure 1) play key sequential roles in the progressive conversion of both enantiomers of the chiral bicyclic monoterpene firstly to the same monocyclic pathway intermediate 2-oxo-∆^3^-4,5,5-trimethylcyclopentenylacetic acid (OTE), and then subsequently to ∆^2,5^-3,4,4-trimethyl- pimelyl-CoA, the first aliphatic pathway metabolite. However, it was over 40 years later [4] before the full significance of the large transmissible genetic element for these monooxygenases and the other enzymes of the camphor degradation pathway was fully appreciated.

Interestingly, the three oxygen-dependent steps in the camphor degradation pathway are each catalysed by a different monooxygenase type. Firstly, the initial activation of each camphor enantiomer to its equivalent chiral bicyclic *exo*-hydroxyketone is catalysed by the same well characterised multicomponent enzyme camphor 5-*exo*-hydroxylase, often referred to by the trivial name cytochrome P450 monooxygenase (cytP450MO: step A, Figure 1). Following subsequent desaturation, again by a single nonselective dehydrogenase, the two resultant diketone enantiomers then undergo oxygen-dependent Baeyer-Villiger-type ring expansion to the corresponding chiral bicyclic lactones (steps C and D, Figure 1), reactions catalysed by the dedicated highly enantioselective enzymes 2,5-diketocamphane 1,2-monooxygenase and 3,6-diketocamphane 1,6-monoxygenase. These are Type II Baeyer-Villiger monooxygenases (BVMOs: [5]), and are often referred to by the trivial name of ketolactonases. The chiral ketolactone products of this second oxygen-dependent step are intrinsically unstable, and undergo spontaneous ring opening and subsequent rearrangement, a chemical sequence which serves to converge the two separate branches of the catabolic pathway by generating the same achiral monocyclic ketoacid OTE. Finally, following prior activation of OTE to its CoA ester, a third oxygen-dependent step (step F, Figure 1) catalysed by 2-oxo-∆^3^-4,5,5-trimethylcyclopentenylacetyl-CoA monooxygenase (OTEMO), a Type I BVMO 5, then generates 5-hydroxy-3,4,4-trimethyl-∆^2^-pimeyl-δ-CoA-lactone. This is another lactone that is chemically unstable, and consequently undergoes spontaneous ring opening and subsequent rearrangement to generate ∆^2,5^-3,4,4-trimethylpimelyl-CoA, thereby completing the transformation of the chiral bicyclic monoterpene into the first aliphatic intermediate in the camphor degradation pathway.

The seminal study of Iwaki et al. [4] confirmed for the first time that along with the requisite genes for all of the other enzymes necessary to catabolise camphor to ∆^2,5^-3,4,4-trimethylpimelyl-CoA, genes coding for each of the characterised monooxygenase-catalysed steps of camphor metabolism are located exclusively within the same 40.45-kb region of the large linear 533-kb CAM plasmid (Figure 2). This includes two very similar copies (*camE*_25-1_ [*orf*4], *camE*_25-2_ [*orf*22]) of the gene coding for 2,5-diketocamphane 1,2-monooxygenase, the highly enantioselective ketolactonase that participates in the catabolism of (+)-camphor, the more widely distributed isomer [6].

Additionally, there are two other putative oxygen-dependent enzymes coded for by *orfs* located within the same 40.45-kb region of the CAM plasmid. However, any potential involvement of these additional oxygen-dependent activities (*orf* 3 = cyclohexanone monooxygenase-like BVMO; *orf* 23 = luciferase-like monooxygenase) in camphor degradation by *P. putida* ATCC 17453 is currently unrecognised.

While camphor 5-*exo*hydroxylase [7,8] and OTEMO [9] have been characterised in considerable biochemical and structural detail, the enantiocomplementary diketocamphane monooxygenases (DKCMOs) have received comparatively little attention. This is in part due to the fact that for over 50 years these Type II BVMOs were mistakenly believed to be flavin prosthetic group-containing flavoproteins (single-component flavoprotein monooxygenases in current terminology [10]) that generate the requisite FMNH_2_ in situ from active site-bound FMN by deploying NADH sourced from another enzyme various referred to NADH oxidase [11] or NADH dehydrogenase [12], and with which the DKCMOs were envisaged as being able to cooperate to form in each case an equivalent loosely bound functional complex. Only comparatively recently [4] has it been established that the DKCMOs are in fact two-component FMN-dependent monooxygenases (fd-TCMOs: [13]). Unlike the more widely distributed Type I BVMOs which are two-component FAD-dependent monooxygenases [14,15], they are dependent on preformed FMNH2 (FNR) as a co-substrate [16] which is transferred by rapid free diffusion from a number of alternative distal donor activities acting as flavin reductases [17]. This fundamental functional misconception was compounded by other relevant misinterpretations (*vide infra*) that were legacies of the research initiated by Gunsalus in the 1960s, and that persisted until corrected many years later [17,18]. The dual aim of the current review is firstly to consolidate both the relevant historical and more recent research, and thereby illustrate the evolution of our present understanding of the enantiocomplementary ketolactonases from *P. putida* ATCC 17453 as archetypal fd-TCMOs, and secondly to catalogue their established roles as biocatalysts of xenobiotic substrates, including their successful deployment in chemoenzymatic synthesis.

## 2. Early Research at the University of Illinois

The impetus to study the microbial catabolism of camphor, from which our current understanding of the DKCMOs of *P. putida* ATCC 17453 has evolved over the last 60 years, was sparked by the interest of Irwin Gunsalus in how compounds such as camphor (‘that little grease-ball’) were rapidly drawn into the carbon cycle by microbial enzymes that could activate dioxygen without risk [19]. These initial studies focussed exclusively on the catabolism of the (+)-isomer of the bicyclic monoterpene. Enrichment selection using (+)-camphor as the sole carbon source was used to isolate *P. putida* ATCC 17453 from the heavily polluted Boneyard Creek on the Urbana campus, and successful outcomes from the research programme initially depended heavily on ‘extraction of the broths at the end of the logarithmic growth phase, and chromatography of the neutral fraction yielded’ [20]. Progress was aided significantly by an established collaboration between Gunsalus and the eminent organic chemist E. J. Corey, and resulted in the correct identification in their initial 1959 publication of not only 2,5-diketocamphane, but also both 5-*exo*- and 5-*endo*hydroxycamphor, OTE, and the free acid form of ∆^2,5^-3,4,4-trimethylpimelyl-CoA as putative pathway intermediates of (+)-camphor catabolism (Figure 1). The obvious chemical relationship between the diketone, its proposed bicyclic bridgehead lactone and hence OTE, plus the additional detection of a relevant monocyclic lactone metabolite promoted a subsequent search for the corresponding lactonizing enzyme(s) in the (+)-camphor-grown bacterium able to promote ‘a succinct process for the cleavage of both carbocyclic rings’ [20]. In turn, this resulted in a seminal report recording the detection in a cell-free extract of the bacterium of an activity which converted 2,5-diketocamphane to OTE only when supplemented with NADH in the presence of oxygen [21]. This represents the first published evidence for a directly assayable activity referred to as a ketolactonase—‘an enzyme system for cyclic ketone lactonization’—this being the ketolactonase currently designated as 2,5-diketocamphane 1,2-monooxygenase (EC 1.14.14.102). Significantly, in a wider context, it also represents the first characterisation of any in vitro enzyme system functioning specifically as a BVMO. Further detailed studies of the metabolites extracted from large volumes of the spent late log phase medium generated by the (+)-camphor-grown *P. putida* ATCC 17453 subsequently confirmed unequivocally the importance of oxygen-dependent lactonization reactions for the successive cleavage of both rings of the bornanone carbocyclic structure [22].

A related preliminary study [23] indicated that growth of *P. putida* ATCC 17453 on (−)-camphor induced a complementary catabolic pathway involving the formation of 3,6-diketocamphane which was subsequently lactonized by an equivalent stereoselective 3,6-diketocamphane 1,6-monooxygenase, although it was only 30 years later that this enzyme was studied in any detail (*vide infra*).

Following the development of a successful ion exchange-based chromatographic separation protocol [24], initial detailed studies were focussed on the (+)-camphor induced ketolactonase enzyme [25]. Neither the amino acid composition or any evidence of subunit structure were investigated. The partially purified enzyme, also referred to as *E*_2_, was reported to be an 80-kDa FMN-dependent monooxygenase containing both nonheme and heme iron as well as bound flavin (Table 1). The chelating agent bipyridyl was demonstrated to have a significant inhibitory effect, which at the time was interpreted to indicate the active involvement of the nonheme iron in the catalytic activity of *E*_2_.

Consequently, the enzyme was deemed to be a flavoprotein containing both tightly bound FMN and iron in the active site of the enzyme, although subsequent studies have confirmed that none of these conclusions are correct (*vide infra*). Further, *E*_2_ was confirmed to function as a monooxygenase in an electron transport complex with a second separate flavoprotein enzyme referred to at the time [25] as FMN-coupled NADH oxidase or *E*_1_, with a reported MW of 50-kDa, although this proved to be a considerable overestimation (*vide infra*). It was suggested that the two participating enzymes are only loosely coupled to form a fragile complex in vivo, with flavin prosthetic group of *E*_2_ acting as sort of a molecular bridge (Figure 3A), and that the complex serves to transfer electrons from NADH sequentially through FMN and iron to diatomic oxygen to generate the activated form of oxygen necessary to promote the lactonization reaction (Figure 3B). Some support for these proposals was obtained when a subsequent modified purification procedure developed by Trudgill et al. [12] succeeded in purifying an active camphor lactonizing *E*_1_*-E*_2_ complex. Again, *E*_2_ was deemed to be a functioning flavoprotein containing ‘several flavin molecules per molecule of complex’. However, subsequent research by Yu and Gunsalus [26] using a further refined separation protocol failed both to detect any iron (heme or nonheme) in purified preparations of the ketolactonase or confirm the chelation-based inhibitory effect of bipyridyl, and consequently the proposed role of ferrous-ferric ion interchange in the functioning of *E*_2_ was no longer considered to be correct, which remains the current consensus position.

Other investigations of *E*_2_ had found that the enzyme could catalyse the lactonization not only of 2,5-diketocamphane but also (+)-camphor, 5-*exo*-, and 5-*endo*hydroxycamphor as well. This outcome, along with the recognition that growth of *P. putida* ATCC 17453 on (+)-camphor also induced reversible 5-*exo*- and 5-*endo*hydroxycamphor dehydrogenases [27] and the characterisation of 1,2-campholide as an early metabolite [28] prompted speculation that the interconversion of the bicyclic terpene to the achiral monocyclic ketoacid OTE by the bacterium may operate in vivo as a form of metabolic grid (Figure 4). However, the respective recorded *K_m_* values suggested that the diketone was the only viable substrate in vivo [29].

Considerable attention was also focused on *E*_1_, the flavoprotein that had already been shown to be able to transfer the reducing power necessary for *E*_2_ to function as a monooxygenase. The enzyme, previously referred to as FMN reductase [21], then FMN-coupled NADH oxidase [25], was renamed again, this time as NADH dehydrogenase [30], and reported to be monomeric on the basis of the detection of a single N-terminal methionine by amino acid analysis. Rather than the earlier reported value of 50-kDa [25], the enzyme was now claimed to have a mean MW of 36-kDa calculated from sedimentation velocity data. However, subsequent research has established unequivocally that it is a 2 × 18.5-kDa homodimer (*vide infra*). In this respect it is significant that Trudgill et al.’s studies of the enzyme predate the development of SDS-PAGE as a technique to investigate the subunit structure of macromolecules [31]. It was shown to contain binding sites for both FMN and FAD, although only FMN was actively involved in transferring reducing power from NADH to *E*_2_. It was speculated that the role of the bound FAD may instead be to enhance the coupling of *E*_1_ and *E*_2_ and hence increase the catalytic efficiency of the loosely bound complex (Figure 3A), although no supporting evidence for such a role was advanced.

The studies of purified preparations of both 2,5-diketocamphane 1,2-monooxygenase (*E*_2_) and NADH oxidase/dehydrogenase (*E*_1_) undertaken at the University of Illinois throughout the 1960s were conducted invariably on enzymes sourced from camphor-grown cells of *P. putida* ATCC 17453 in the late log-early stationary phase of growth. However, although it was recognised at the time that *E*_1_ and *E*_2_, like a number of other enzymes of the camphor degradation pathway were subject to regulatory control, and consequently exhibited different titres at the various stages throughout the growth cycle on camphor-based minimal medium [11,28,29,32], this phenomenon was not examined further. The significance of this omission is that it is now known (*vide infra*) that both the nature and number of the enzymes that can serve as suppliers of reducing power to 2,5-diketocamphane 1,2-monooxygenase differ throughout the different phases of camphor-dependent growth, and that Conrad et al.’s *E*_1_ [25,30] may in fact be an enzyme only relevant for post-growth stationary phase secondary metabolism.

## 3. Post-Gunsalus Research at the University of Aberystwyth

Following cessation of the Gunsalus-led decade of pioneering research on the DKCMOs induced in camphor-grown *P. putida* ATCC 17453, it was not until Peter Trudgill, a former collaborator of Irwin Gunsalus, reopened camphor-based studies as a part of a broader research initiative to investigate terpene metabolism by bacteria [33] that any further progress was made in characterising these fascinating lactonizing enzymes. No doubt reflecting Trudgill’s formative apprenticeship, these studies, like those previously undertaken at the University of Illinois, were undertaken exclusively on enzymes sourced from camphor-grown cells of *P. putida* ATCC 17453 harvested in the late log-early stationary phase of growth, the significance of which has only become apparent with recent more extensive investigations (*vide infra*).

Two key initiatives promoted those advances that were made [34]. These were firstly the development of improved purification protocols, and secondly the deployment of methods to characterise the quaternary structure of the resultant purified proteins. Purification of the oxygenating moiety of 2,5-diketocamphane 1,2-monooxygenase [34] by a combination of ammonium sulfate fractionation followed by successive ion exchange and affinity chromatographic steps yielded a preparation progressively purified 71.2-fold with a MW of 78-kDa, which corresponded well with the previous estimated MW of 80-kDa for *E*_2_. Additionally, for the first time, it was confirmed to be a homodimer, being separable by SDS-PAGE into two apparently identical subunits of equal MW. Electrophoretic separation of the native protein yielded two distinct isoenzymic activities, referred to as A and B, both of which had very similar MWs and the same subunit composition. Although not recognised as such, instead being postulated at the time to represent post-translational modification of a single protein type, this did provide the first indirect evidence for the existence of two different genes on the CAM plasmid coding for isoenzymic forms of the monooxygenase, subsequently identified several years later by Iwaki et al. [4] as *camE*_25-1_ (*orf* 4) and *camE*_25-2_ (*orf*22). Another relevant outcome from studying the new highly purified monooxygenase preparation was to confirm the absence of iron (Fe^3+^) at a detectible level, as previously proposed by Yu and Gunsalus [26].

A later study [6] then refined the purification protocol further by introducing a Q Sepharose-based FPLC final step, providing additional access to homogeneous preparations of (−)-camphor-induced 3,6-diketocamphane 1,6-monooxygenase, which was confirmed to be another homodimer. The MW of the purified native enzyme was determined to be 76-kDa by ultracentrifugal analysis, characterising it as the smaller ketolactonase. Notably, these data do not correspond either with direct studies of these two recombinantly expressed enzymes [35,36], or the equivalent calculated outcomes from respective amino acid analyses made possible by the sequencing of the corresponding genes on the CAM plasmid [4]; both of these analyses confirm that 3,6-diketocamphane 1,6-monooxygenase is the larger homodimeric enzyme (2 × 42.3-kDa compared 2 × 40.5-kDa).

Comparative growth studies with the separate enantiomer of camphor demonstrated that they each induced significant titres of both 2,5-DKCMO and 3,6-DKCMO, a cross-inducibility outcome which contrasts with the extremely high enantioselective catalytic activity recorded towards only the camphor enantiomer from the corresponding chiral series when highly purified homogeneous preparations of the two enzymes were tested. Later more comprehensive programmes of relevant research have consistently confirmed this striking contrast between the mutually enantioexclusive aspect of the catalytic activity of the ketolactonases [4,37,38,39], and the significant levels of cross-inducibility promoted by both camphor enantiomers [17,37,40]. It is possible that the relatively small variations in the extent of cross-inducibility recorded in the separate studies may reflect differences in the relative purity of the respective samples of camphor used, which in each case were commodity grade chemicals.

Summarising a decade of studies at the University of Aberystwyth, Jones et al. [6] reiterated the proposal first made in the 1960s that both ketolactonases are flavoproteins with bound FMN as the prosthetic group, and that each forms a loosely associated active oxygenating complex with an equimolar amount of the 36-kDa monomeric NADH oxidase/dehydrogenase previously designated *E*_1_ [25]. These fundamentally incorrect conclusions remained unchallenged for another 20 years before the DKCMOs were finally correctly identified and reclassified as fd-TCMOs that function by accepting pre-reduced FMNH_2_ from a number of different remotely located flavin reductases.

## 4. A New Millennium and a New Perspective—The Ketolactonases as fd-TCMOs

Although applications of the ketolactonases as biocatalysts to generate synthons of value for chemoenzymatic synthesis were investigated at the University of Exeter throughout the 1990s as a part of a more extensive BVMO-focussed programme (*vide infra*), mode of action studies were not included, and it is only in the last decade that new initiatives resulting in novel and unexpected outcomes have led to a reappraisal of basic functional aspects of the biochemistry of these enzymes. Consequently, it is now accepted that rather than being simple flavoproteins, a proposal first made in the mid-1960s [11,24] and serially reiterated subsequently, the ketolactonases are actually members of the FMN-dependent two-component monooxygenase family (fd-TCMO; [13]). Thus while conventional flavoproteins function with one or more flavin prosthetic groups that are covalently bound in the active site and reduced in situ, fd-TCMOs (EC 1.14.14.x) utilise a pre-reduced flavin cofactor as a substrate, sourced from the cooperative activity of a physically separate flavin reductase (EC 1.5.1. 36-40). All such enzymes characterised to date have been discovered in bacteria, and while most oxygenate carbon atoms, some such as DszC from the thermophilic bacterium *Paenibacillus* sp. strain A11-2 [41] can function with heterocyclic sulfur atoms.

The first indication that the ketolactonases of *P. putida* ATCC17453 are fd-TCMOs rather than simple flavoproteins resulted from an initiative by Uwe Bornsheuer at the University of Greifswald [35] to expand the use of these enzymes as biocatalysts in chemoenzymatic synthesis. The then-conventional procedures to obtain the enzymes only yielded relatively small amounts of pure material from large amounts of camphor-grown biomass [6,40]. Consequently, a research programme was initiated to generate increased amounts of 2,5-diketocamphane 1,2-monooxygenase by recombinant expression in *E. coli* BL21(DE3) of a relevant gene isolated from plasmid DNA with the aid primers derived from GenBank entry AY450285.1 (2003), and now known to correspond to *camE*_25-1_ [4]. Although only the gene coding for the oxygenating moiety was expressed in the heterologous host, significant lactonizing activity was recorded with partially purified preparations of the recovered recombinant enzyme. Equivalent outcomes were recorded when the gene for 3,6-diketocamphane 1,6-monooxygenase, having first been identified on the CAM plasmid by gene walking PCR from *camE*_25-1_, was then heterologously expressed in the same host [36]. These data were interpreted as indicating that one or more native activities of the *E. coli* host can serve as a flavin reductase to donate FMNH_2_ to the cloned ketolactonases. This unexpected outcome in turn led to the initial recognition of the possibility that the ketolactonases from *P. putida* ATCC 17453 could be fd-TCMOs rather than simple flavoproteins. Several other bacterial fd-TCMOs have been described previously [13], including the luciferases from *Vibrio harveyi*, *V. fischeri*, *Photobacterium phosphoreum*, *P. legionathi*, and *Xenorhahdus luminescens* [42]. Interestingly, bacterial luciferases and the ketolactonases have been grouped together before, as so-called Type II BVMOs, on the basis of a number of other shared functional properties [5]. A dual initiative was then launched [43] to identify both the nature of the competent FMNH_2_-generating enzyme(s) in *E. coli*, and any equivalent enzyme(s) in *P. putida* ATCC 17453 corresponding to *E*_1_, Trudgill et al.’s [6,12,30,34] apocryphal monomeric 36-kDa NADH oxidase/dehydrogenase, ironically an activity transiently referred to as ‘FMN reductase’ [21]. The native *E. coli* activity was successfully identified as the known monomeric 26.2-kDa FMN-dependent flavin reductase Fre [44]. However, while a number of putative flavin reductases were identified in the genome of *P. putida* ATCC 17453 by CODEHOP PCR [41,45], ultimately the latter programme of research was stymied by the chosen search parameters, and proved unsuccessful. Further progress on identifying competent native flavin reductase activities in the pseudomonad then had to await the subsequent inputs of Iwaki et al. [4] and Willetts and Kelly [17,18].

That the ketolactonases from camphor-grown *P. putida* ATCC 17453 are indeed fd-TCMOs was confirmed definitively by Iwaki et al.’s seminal paper published in 2013 [4], which established the true nature of the activity previously misidentified (*vide supra)* as a monomeric 36-kDa protein variously referred to as *E*_1_ [21,24], NADH oxidase [25,34], and NADH dehydrogenase [6,12,30]. This enzyme, now given the trivial name Fred, was conclusively confirmed to be a homodimeric 37-kDa flavin reductase. Samples of Fred prepared from late log-early stationary phase (+)-camphor-grown cells of *P. putida* ATCC 17453 were sequenced, and that data then used to design probes that confirmed that the corresponding gene was located on the chromosomal DNA. Analysis of the gene predicted a corresponding protein product of 170 amino acids, including the presence of the characteristic flavin reductase motifs GDH [46] and YGG [47]. The predicted *M_r_* of the protein was 18,466. Subsequent cloning and overexpression of the gene in *E. coli* BL21(DE3) generated sufficiently large quantities of Fred for the activity to be extensively characterised. Studies with highly purified samples confirmed that the active form of the enzyme was an NADH-specific 2 × 18.5-kDa homodimer that accepts flavins as substrates, with FMN clearly being the preferred option (FMN [*K_m_* = 3.6 μM, *k_cat_* = 283 sec^−1^]: FAD [*K_m_* = 19 μM, *k_cat_* = 128 sec^−1^]). It thus corresponds to a Class II nonflavoprotein reductase [48], which operate a sequential rather than a ping-pong reaction mechanism [13,16]. It appears to most closely resemble NTA-MoB, the flavin reductase component of nitrilotriacetate monooxygenase, an fd-TCMO isolated from *Mycobacterium thermoresistible* [49].

Using a combination of standard cloning and sequencing techniques, Iwaki et al. [4] also undertook a comprehensive study of the CAM plasmid that specifically included a search for genes coding for the oxygenating subunits of the ketolactonases. Contrary to previous predictions [2], the plasmid proved to be a large linear 533-kb double-stranded transmissible genetic element. The search for individual genes not only located the corresponding gene for the oxygenating moiety of 3,6-diketocamphane 1,6-monooxygenase (*camE*_36_) within a contiguous 40.5-kb region of the established *camRDCAB* locus of the CAM plasmid, but also confirmed the isoenzymic nature of the enantiocomplementary 2.5-diketocamphane 1,2-monooxygenase activity by identifying genes corresponding both to 2,5-DKCMO-1 (*camE*_25-1_) and 2,5-DKCMO-2 (*camE*_25-2_) within the same locus (Figure 2). This in turn could explain the two previously uncharacterised electrophoretically separable forms of the enzyme previously reported by Taylor and Trudgill [34] as A and B (*vide supra*).

The confirmation by Iwaki et al. [4] that the CAM plasmid-coded DKCMOs from *P. putida* ATCC 17453 function as fd-TCMOs in cooperation with the genomic DNA-coded homodimeric flavin reductase Fred prompted a subsequent study to investigate whether any other indigenous reductases can support the ketolactonases [17]. This search was promoted by the fact that apart from Fred, all other activities necessary to catabolise camphor to ∆^2,5^-3,4,4-trimethylpimelyl-CoA are coded for by one or more identified genes within the *camRDCAB* locus of the CAM plasmid. Studies of total flavin reductase activity throughout the growth of *P. putida* ATCC 17453 on either succinate- or (+)- or (−)-camphor- based minimal media confirmed that similar titres of activity were detectible in the earliest sampled cells grown on all three media, but that whereas that level remained remarkably consistent in succinate-grown cells throughout all subsequent stages of growth, the flavin reductase titre of cells grown on either (+)- or (−)-camphor-based minimal medium was progressively induced to a maximum 2.8-fold higher level before plateauing after entry into the late exponential-stationary phase of growth (Figure 5). A combination of gel-filtration chromatography, native PAGE, and SDS-PAGE electrophoresis, and comparative sequence data analysis of purified recovered samples confirmed that two monomeric reductases (Frp1, 27.5-kDa and Frp2, 28.5-kDa), corresponding with the previously characterised ferric reductases FprA and FprB from *P. putida* KT2440 [50], were the only relevant activities recorded after growth on succinate. Equivalent constitutive titres of Frp1 and Frp2 were also confirmed after growth on either camphor enantiomer. Similar analysis of the significant additional camphor-induced reductase activity detected two principal additional proteins induced by growth on either terpene enantiomer. One of these was confirmed to be Fred, the 37-kDa homodimer reported previously as the sole reductase activity of *P. putida* ATCC 17453 [4]. The purified monomeric protein responsible for the second inducible flavin reductase activity had an N-terminal amino acid sequence consistent with that of putidaredoxin reductase (PdR), a 48.5-kDa protein coded for by the *camA* gene of the CAM plasmid and known to serve as one of the functioning components of camphor 5-*exo*hydroxylase [51], but which has not been reported previously to perform a similar role with the DKCMOs. Purified samples of the various identified flavin reductase activities (Fred, PdR, FprA plus FprB) were confirmed to be equally active in supporting lactonization reactions with a high degree of enantioselectivity by purified preparations of both DKCMOs.

The relative roles of the each of the identified flavin reductase activities in supporting lactonizing activity by the DKCMOs were then examined in more detail in a subsequent study [18], which both logged the titres of all relevant reductase and oxygenating activities promoted by camphor as a growth substrate, and additionally characterised relevant kinetic data for the generation of FMNH_2_ and its subsequent transfer to the oxygenating subunits of the DKCMOs. The camphor-promoted events were monitored (Figure 6) by following up on the early observation by Gunsalus et al. [29] that during the diauxic growth of *P. putida* ATCC 17453 on a defined succinate plus (*rac*)-camphor medium, the enzymes of the camphor degradation pathway were only expressed after complete exhaustion of the succinate.

With respect to the oxygenating activities, both the timing and extent of induction of 2,5-diketocamphane 1,2-monooxygenase and camphor 5-*exo*hydroxylase were very similar in response to the swop to (*rac*)-camphor as the growth substrate. While the enantiocomplementary 3,6-diketocamphane 1,6-monooxygenase followed a very similar time course, the level of induction was consistently lower, confirming some earlier preliminary observations [37,39,40]. Not unexpectedly, the profile of induction of PdR activity coincided exactly with that of the composite camphor 5-*exo*hydroxylase, of which the monomeric 48.5-kDa protein is a confirmed functional subunit [52]. Kinetic data derived from characteristic changes in the relevant absorption spectra confirmed conclusively that the highly purified PDR preparation accepted reducing power exclusively from NADH thereby generating bound FADH_2,_ which in turn was able to pass reducing power to unbound FMN added subsequently as a cofactor. This sequential transfer of reducing power from NADH to FMNH_2_ explains the confirmed ability of highly purified PdR to function as the effective flavin reductase partner for both enantioselective ketolactonases when functioning as fd-TCMOs [17]. This in itself is significant because it completes the biochemical autonomy of the CAM plasmid as it is now possible to assign plasmid-coded proteins acting as enzymes and/or redox intermediates for every activity necessary to metabolise both camphor enantiomers to the level of ∆^2,5^-3,4,4-trimethylpimelyl-CoA (Figure 1). However, how the available pool of PdR-FADH_2_ is effectively divided in vivo to support two very different outcomes in camphor-grown *P. putida* ATCC 17453 remains to be established.

Detailed investigation of Fred, the 37-kDa homodimeric flavin reductase claimed by Iwaki et al. [4] to correspond to the *E*_1_ activity initially reported as an NADH oxidase or dehydrogenase in early studies [6,25,30,34], confirmed that it is, like PdR, a camphor-induced enzyme. Kinetic studies using a highly purified enzyme preparation confirmed that Fred can only effectively transfer reducing power from NADH to FMN (*K_m_* FMN = 4.2 μM (3.6 μM, Iwaki et al. [4]), *k_cat_* FMN = 294 s^−1^ (283 sec^−1^, Iwaki et al. [4]); *K_m_* NADH = 29 μM), and relevant corresponding double reciprocal plots confirmed that Fred generates FMNH_2_ by a sequential reaction mechanism.

The titres of Frp1 and Frp2 were remarkably consistent throughout each successive stage of diauxic growth, confirming the outcome of the earlier growth study [17] that both Frp1 and Frp2 are constitutively expressed activities in *P. putida* ATCC 17453. Kinetic studies confirmed that a purified preparation of Frp1, while totally inactive with NADPH, was able to serve as a comparatively effective means of generating FMNH_2_ from NADH (*K_m_* = 23 μM) from FMN (*K_m_* = 6.9 μM, *k_cat_* = 310 s^−1^) while showing minimal and no activity with FAD and riboflavin respectively. While Frp2 gave similar outcomes, there was some albeit much lower generation of FMNH_2_ from NADPH (*K_m_* = 92 μM), although the kinetic data suggests that this level of activity is probably not physiologically insignificant. Double reciprocal plots of relevant data confirmed that both Frp1 and Frp2 are EC 1.5.1.x-type Class II non-flavoprotein reductases [44,48] that generate FMNH_2_ by a sequential reaction mechanism [13,16].

The relative contribution of the different assayed FMNH_2_-generating enzymes to the total flavin reductase activity titre throughout the various phases of the post-diauxic camphor-dependent growth of *P. putida* ATCC 17453 (Figure 7) indicated that the initially predominant importance of the combined constitutive Frp1 and Frp2 activities was progressively diminished throughout the early and mid log phases of growth by the steady progressive induction of the CAM plasmid-coded PdR activity. Thereafter, these flavin reductase activities assumed approximately equal importance at the mid log-late log interphase in the growth cycle, before the relative importance of PdR declined as induction of the protein is down-regulated in the later stages of the growth cycle. Significantly, the titre of genome-coded camphor-induced Fred only assumed any importance during the late log phase of growth, progressing to become the predominant FMNH_2_-generating activity after entry into stationary phase, a profile matched concomitantly by declining titres for PdR and both DKCMOs. This time course for the up-regulation of Fred is more consistent with that of an enzyme involved in secondary rather than primary metabolism [53]. It is relevant in this respect, species of *P. putida* are known to produce a wide range of indigenous secondary metabolites [54], including various polyketides [55], for which fd-TCMOs are known to play important roles as ‘tailoring enzymes’ [56,57,58].

That both the growth substrate-dependent and growth phase-dependent changes in the titres of the DKCMOs and the flavin reductases Fred and PdR are illustrative of the wider importance of transcriptional control in up-regulating the pathway for the degradation of alicyclic camphor to aliphatic ∆^2,5^-3,4,4-trimethylpimelyl-CoA has been confirmed by recent research using the relevant inhibitors rifampicin and actinomycin D [59]. This comprehensive study monitored the differential rates of synthesis of both genome-coded Fred and a number of CAM plasmid-coded activities including the ketolactonases in response to both camphor and key degradation pathway intermediates to establish the various induction and repression circuits that regulate relevant activities in *P. putida* ATCC 17453 (Figure 8). Augmented by the outcomes from some relevant earlier studies [6,17,25,30], the results confirmed that the genes that code for the enantioselective DKCMOs are subject to induction by the corresponding camphor enantiomer, along with the *camRDCAB* polycistronic operon that codes for camphor 5-*exo*hydroxylase and 5-*exo*-hydroxycamphor dehydrogenase. This coordinate transcriptional control of the first three successive steps of the catabolic pathway by the initial substrate represents ‘from the top’ coordinate pathway regulation, and has been reported in a number of other catabolic pathways in other *Pseudomonas* spp. [60,61]. Further, the relevant differential rates of synthesis demonstrated that each enantioselective ketolactonase as well as being induced by its own corresponding diketone substrate, was cross-induced by the complementary chiral diketocamphane from the opposite enantiomeric series, and additionally back-induced by OTE. These two interesting forms of induction confirm cross-inducibility and product induction respectively as two additional important elements of transcriptional regulatory control of the ketolactonases in camphor-grown *P. putida* ATCC 17453. Product induction or so-called ‘from the bottom’ regulation is not unique to the DKCMOs of camphor-grown *P. putida* ATCC 17453. It has been characterised in a number of other bacterial catabolic pathways [62,63,64,65,66], and has been speculated to reflect the evolution of catabolic pathways ‘from the bottom to the top’ by the sequential acquisition of additional units of physiological function [60,61]. The confirmation of cross-inducibility of the DKCMOs by their complementary chiral diketocamphane pathway intermediates supports earlier studies of the camphor degradation pathway which have consistently reported an equivalent significant element of cross-inducibility of both ketolactonases by each enantiomer of camphor [6,17,37,40]. The broad specificity of the relevant repressor proteins implicated by these various transcriptional controls, allied to the established patterns of coordinate induction has been suggested as illustrative of a more general phenomenon characteristic of a number of different catabolic pathways of pseudomonads [11,29], thereby vesting individual species with their acknowledged impressive metabolic versatility as illustrated by *P. putida* KT2440 [67].

## 5. Structural Studies of the Ketolactonases

It was the seminal investigation by Iwaki et al. [4] which successfully sequenced a contiguous 40.5-kb region of the established *camRDCAB* locus of the CAM plasmid that, by locating the corresponding genes, presented the first opportunity to compare the complete primary structure of the ketolactonases induced in camphor-grown *P. putida* ATCC 17453. As well as locating the corresponding gene for the oxygenating moiety of 3,6-diketo-camphane 1,6-monooxygenase (*camE*_36_), an unexpected outcome of this study was confirmation of the isoenzymic nature of the enantiocomplementary 2,5-diketocamphane 1,2-monooxygenase activity by identifying two corresponding genes (*camE*_25-1_ and *camE*_25-2_) on the CAM plasmid (Figure 2). Contrary to the previous proposal made by Jones et al. [6], translation of the relevant nucleotide sequences confirmed that 3,6-diketocamphane 1,6-monooxygenase is the larger protein. It comprises 378 amino acid residues compared to the 363 amino acid residues that characterise the two proteins coded for by the *camE*_25-1_ and *camE*_25-2_ genes. Standard alignment tools serve to emphasise the considerable similarity between all the three sequences (Figure 9). While the two isoenzymic 2,5-diketocamphane 1,2-monooxygenases exhibit the highest alignment score (1852 using BLOSUM62), the alignment scores for both isoenzymes when compared to 3,6-diketocamphane 1,6-monooxygenase (961 and 965) are very similar.

Surprisingly little other research has been undertaken on comparative structural features of the various DKCMOs induced in camphor-grown *P. putida* ATCC 17453. The only study that directly compares 2,5-diketocamphane 1,2-monooxygenase and 3,6-diketocamphane 1,6-monooxygenase deployed a ‘substrate mapping’ analysis [68] of their respective active sites based on the outcomes obtained from using highly purified preparations of the enzymes to biotransform twenty-three organosulfide substrates for which the absolute configuration of the resultant sulfoxides could be confirmed [39]. Because both ketolactonases acting as fd-TCMOs can undertake molecular oxygen plus FMNH_2_-dependent electrophilic oxygenation of competent organosulfide substrates in their respective active sites thereby generating one or both corresponding chiral sulfoxides, the resultant outcomes define the stereogenic face(s) from which the ‘active oxygen’ must have been delivered. These results in turn enabled so-called ‘cubic space models’ [69] of the active sites of the enantiocomplementary ketolactonases to be constructed and compared, with each model being composed of four subdomains (H1 = pocket for dialkyl moiety of (*S*)-dialkyl sulfoxides and the aromatic moiety of (*S*)-alkyl aryl sulfoxides, H2 = equivalent pocket to H1, but for (*R*)-sulfoxides, H3 and H4 = pockets for the alkyl moiety of (*S*)- and (*R*)-alkyl aryl sulfoxides respectively). While the resultant H1 and H4 subdomains of both enzymes exhibit very similar dimensions, the H3 subdomain of 2,5-diketocamphane 1,2-monooxygenase was bigger than the equivalent subdomain of 3,6-diketocamphane 1,6-monooxygenase. However, this was more than compensated for by the relative sizes of the H4 subdomains, which was substantially larger in 3,6-diketocamphane 1,6-monooxygenase. By superimposing the relevant four subdomains for each ketolactonase along the S=O axis for each competent substrate, composite models of the two respective active sites were constructed (Figure 10). The models emphasised that the active site of the 3,6-diketocamphane 1,6-monooxygenase was significantly larger than that of the enantiocomplementary ketolactonase. This in turn may explain why this enzyme exhibites lower enantio- and regioselectivity in undertaking both nucleophilic (ketones to lactones) and electrophilic (sulfides to sulfoxides) oxygenations than 2,5-DKCMO, as illustrated by the respective outcomes recorded with both the 7-*endo*methyl- and 7,7-dimethyl-substituted bicyclo[3.2.0]hept-2-en-6-ones, and methyl-*para*-tolyl sulphide (Table 2). While giving meaningful insights into aspects of substrate binding and the comparative selectivity of the two ketolactones, the study provided no insights into other crucial active site-located functional activities including the binding of FMNH_2,_ and its critical role in generating ‘active oxygen’ by these fd-TCMOs.

Given the general caveat that ‘any attempt of predicting the selectivity of an enzymatic reaction based on X-ray data is comparable to explaining the complex movements in a somersault from a single photographic image’ [70], efforts to determine a crystal structure for either ketolactonase, and thereby enable successful modelling of the crucial flavin-dependent oxygenating activities within the active site, have proved to be problematical. The clearly confirmed finding that BVMOs [71], like other enzymes [72], undergo sequential structural dynamics during catalysis only serves to exacerbate these issues. Initial attempts to generate X-ray diffraction-grade crystals of 2,5-diketocamphane 1,2-monooxygenase were unsuccessful [73], possibly because the purified preparation used contained a mixture of both isoenzymic forms of the enzyme, and no further studies have been reported. Conversely, the enatiocomplementary ketolactonase generated multiple crystal forms [73,74], the best of which belonged to the orthorhombic space group *P*2_1_2_1_2_1_ and diffracted to beyond 2 Å resolution, allowing a complete data set to be collected to 2.5 Å resolution at 100 K, but which defied refinement. Sometime later the data was re-examined by Isupov and Lebedev [75] using a molecular replacement (MR) phasing technique that deployed both the α- and β-subunits of the luciferase from *Vibrio harveyi*, a characterised α/β-heterodimeric fd-TCMO in which both subunits share low (16%) sequence homology with the homodimeric ketolactonase. However, MR failed at the cross rotation function step, and consequently a model structure was generated employing a noncrystallographic symmetry-constrained exhaustive search for which a synthetic α_2_-dimer of the luciferase was considered a better search model. The synthetic α_2_-dimer itself was constructed from a 2.4 Å resolution 3D structure [76] which predates recognition that the β-subunit of the luciferase, although not catalytically active, has a profound allosteric effect on the 3D shape of the α-subunit [77]. Most significantly, the mobile loop adjacent to the active site that promotes the relevant allosteric pathway was crystallgraphically disordered in Fisher et al.’s 1995 study [76]. The subsequent development of a successful overexpression system for *camE*_36_ by Kadow et al. [36], providing access to significant amounts of the purified ketolactonase, then led Isupov et al. to renew interest in resolving the crystal structure of the ketolactonase [78]. However, while the outcome (PDB code 4UWM) had a reported high resolution (1.9 Å and low *R*-values), it remained dependent on an MR step deploying the same problematic synthetic α_2_-dimer of luciferase that was used in the earlier study [75].

Being an fd-TCMO for which FMNH_2_ is a co-substrate [4], the ketolactonase binds FMN more than 500-fold less effectively [18], which stymied attempts to assign the orientation of the oxidised flavin in a co-crystallised complex [78]. Isupov et al.’s alternative flavin-based modelling studies were detrimentally influenced by the same relative binding kinetics, but additionally were compromised by the inherent issue of the three-dimentional shape of the relevant flavin co-substrate; thus while the isoalloxazine ring of FMN is planar, the equivalent structural unit of FMNH_2_ can arc along the N5-N10 axis by up to 144° [79]. Consequently, the locations of N5 and the heterocyclic moiety of this tricyclic ring, which both serve as key determinants of functional oxygenating activity, are ill-defined [80]. A better understanding of the key biochemical events involved in the generation and deployment of ‘active oxygen’ by the DKCMOs of camphor-grown *P. putida* ATCC 17453 acting as lactonizing fd-TCMOs must await further more relevant study.

## 6. Chemo-, Regio-, and Enantioselectivity of the Ketolactonases: Natural and Xenobiotic Substrates, Plus Biocatalytic Applications

The ketolactonase-catalysed biooxidation of a ketone to lactone corresponds to the equivalent peracid-catalysed Baeyer-Villiger chemical oxidation, a reliable method of preparing esters and lactones first reported in 1899 [81]. While examples are known of metal (II) chiral complexes able to catalyse asymmetric chemical lactonizations [82], almost all reported chemical Baeyer-Villiger oxidations are not enantioselective, and the regiochemistry of oxygen insertion strongly favours a single major product. Significantly, the ketolactonases of camphor-grown *P. putida* ATCC 17453 exhibit very different regio- and enantiospecific lactonization characteristics as exemplified by the contrasting outcomes recorded from the chemical oxidation and the enantiodivergent biooxidation of (*rac*)-bicyclo[3.2.0]hept-2-en-6-one by metachlorperbenzoic acid and 2,5-DKCMO respectively (Figure 11), a result that serves to illustrate the growing importance of biocatalysis in organic chemistry and biotechnology [83,84].

The acknowledged natural substrates of the enantiocomplementary 2,5- and 3,6-DKCMOs are 1,7,7-trimethylbicyclo[2.2.1]heptan-2,5-dione (2,5-bornandione, 2,5-diketocamphane) and 1,7,7-trimethylbicyclo[2.2.1]heptan-3,6-dione (3,6-bornandione, 3,6-diketocamphane) respectively. The only known occurrence of these diketones in Nature is as intermediates in the dedicated pathway for the degradation of the corresponding (+)- and (−)-enantiomers of camphor (Figure 1). Camphor itself is a relatively rare bicyclic monoterpene, both isomers of which are found principally in the wood of some members of the laurel family, most notably *Cinnamomum camphora* (the camphor laurel tree), and *Ocotea usambarensis*, and also in significant amounts (5–10%) in the oils produced by *Rosemarinus offinalis* and *Crysanthemum sinense*. While some early research with partially purified 2,5-DKCMO [29] reported that (+)-camphor, and to a significantly lesser extent 5-*exo*- and 5-*endo*hydroxycamphor each showed some limited activity as poor alternative substrates, as reflected by their very much higher *K_m_* values relative to 2,5-diketocamphane (*vide supra*, Figure 4), other more recent chemoselectivity studies have established that a number of other naturally occurring carbonyl-substituted isoprenoids (pulegone, carvone, thujone, nopinone) including the structurally-related monoterpene fenchone are completely inactive with the enzyme [4,33].

With the viable natural substrates, highly purified preparations of both the native and recombinantly expressed forms of 2,5-DKCMO and 3,6-DKCMO exhibit absolute stereoselectivity for the corresponding diketocamphane and camphor isomers from the corresponding enantiomeric series (Table 3A, [4,6,17,37,38,40]); the unstable nature of the lactone products resulting from the diketocamphane enantiomers is consistent with the reported absence of any detectible pertinent lactone hydrolase activity in (*rac*)-camphor-grown *P. putida* ATCC 17453 [33]. These aspects of absolute stereoselectivity of the DKCMOs contrasts firstly with the known lack of stereoselectivity of both camphor 5-*exo*hydroxylase [28] and 5-*exo*hydroxycamphor dehydrogenase [85], the initiating and immediately preceding enzymes respectively in the camphor degradation pathway, and secondly with the fact that both the unstable lactones formed by the enantio-complementary DKCMOs then spontaneously rearrange to the same achiral metabolite OTE (Figure 12). Significantly, any implied potential of *P. putida* ATCC 17453 to discriminate between (+)- and (−)-camphor as a growth substrate by selectively up-regulating the corresponding stereoselective ketolactonase (Figure 12) is negated by a combination of the established cross-inducibility of the relevant CAM plasmid genes by each of the enantiocomplementary camphor and diketocamphane isomers, and the multivalent product induction of the genes by the shared degradation pathway intermediate OTE (*vide supra*, [59]). The very similar nucleotide sequences of the *camE*_25-1_, *camE*_25-2_, and *camE*_36_ genes on the CAM plasmid (*vide supra*) suggests that they probably arose by gene duplication and subsequent divergence [86], although any evolutionary advantage of the isoenzymic 2,5-DKCMOs relative to the implicit additional genetic load is difficult to imagine given both the identical outcome reported for highly purified preparations of both recombinantly expressed isoenzymes with the natural substrate (+)-camphor [4], and the reported rarity of the bicyclic monoterpene as a natural substrate in the biosphere [6,33]. It may be significant that the *camE*_25-1_ and *camE*_25-2_ genes are located on opposite strands of the CAM plasmid [4], and consequently may possibly be divergently transcribed, although this was not evident from a recent study of transcriptional control of the CAM plasmid-coded enzymes [59]. Interestingly, a comprehensive review has concluded that enantiocomplementary and duplicate enzymes are surprisingly common in Nature, and may in some cases arise serendipitously [87].

The number of xenobiotic compounds tested as potential substrates for one or both of the DKCMOs is considerable, and includes both carbonyl-containing compounds that can undergo nucleophilic oxygenation and organosulfides that can undergo electrophilic oxygenation. Amalgamated summaries of the characterised outcomes recorded when substrates (both xenobiotic and natural) have been used specifically to challenge various highly purified preparations of the native and recombinant ketolactonases are presented in Table 3A (bicyclic ketones), Table 3B (other alicyclic ketones), Table 3C (aliphatic ketones), and Table 3D (organosulfides). Preliminary data obtained with crude cell-free extracts or incompletely separated enzyme preparations containing mixtures of both the 2,5- and 3,6-DKCMOs are not included. These data have significance in a number of different contexts.

The established chemoselectivity of both ketolactonases in performing nucleophilic oxygenations of xenobiotic compounds is confined almost exclusively to the lactonization of alicyclic ketones. One apparent exception is the recombinantly expressed 3,6-DKCMO generated and tested by Kadow et al. [36] that was reported to yield uncharacterised ester products from acetophenone, 4-phenyl-2-butanone, and *n*-decan-2-one. However, this particular cloned enzyme is characterised by very low catalytic activity and the apparent substrate specificity, which differs from that of both Iwaki et al.’s equivalent preparation [4] and the corresponding native enzyme [6,17,37,38,39,40] in other respects (*vide supra*), may just represent background “noise”. Interestingly, however, another two-component FMN-dependent monooxygenase, the luciferase from *Photobacterium phosphoreum* NCIMB 844, can biotransform both its natural substrate *n*-dodecanal to the equivalent carboxylic acid, and the abiotic ketone 2-tridecanone to methyl dodecanoate [88]. In comparison, the chemoselectivity of both DKCMO activities in undertaking electrophilic oxygenations is wider, and competent substrates include not only dialkyl sulphides, but also alkyl aryl sulphides, alkyl benzyl sulphides, and bicyclic sulphides, an attribute that has been exploited to build cubic space models of the active sites of the ketolactonases (*vide supra*).

Although with hindsight the potential importance for the biotransformation of xenobiotic substrates by the enantiocomplementary ketolactonases was evident from the studies undertaken at the University of Illinois in the 1960s, this was not exploited until developed as a key part of the pioneering interdisciplinary research undertaken at the University of Exeter under the direction of Prof Stanley Roberts in the 1990s [5]. The initial trigger to examine the capacity of the ketolactonases as biocatalysts to generate valuable synthons for chemoenzymatic synthesis stemmed from the recognition [89] that the established outcomes of the metabolism of 2,5- and 3,6-diketocamphane by these enzymes [3,6,33] can in principle be exploited to promote equivalent biotransformations of even the simplest (*rac*)-bicyclo[2.2.1]ketone norbornan-2-one to produce bridgehead lactones in opposite chiral series which, when combined with an appropriate subsequent ring-opening step, should yield two enantiocomplementary cyclopentanes replete with two chiral centres (Figure 13A). The judicious choice of (*rac*)-ketone substrates with a suitable pattern of substitution should in principle result in enantiocomplementary lactones which on ring-opening yield two equivalent cyclopentanes replete with four chiral centres (Figure 13B). By the alternative deployment of either a chemical (lithium aluminium hydride) or enzymatic (lactone hydrolase) ring-opening step, the number of chiral products can again in principle be doubled (Figure 13B). Such highly substituted small carbocyclic molecules are versatile synthons for incorporation in key steps in the chemoenzymatic synthesis of commercially valuable products [90]. In practice, the unexpected results consistently obtained when highly purified preparations of the ketolactonases were challenged with various (*rac*)-bicyclo[2.2.1]heptan-2-ones proved surprising. When the unsubstituted bicyclic terpene (*rac*)-norbornan-2-one (norcamphor) was tested [91], both ketolactonases proved to be stereochemically congruent by generating the same bridgehead (1*R*,5S)-lactone, albeit 3,6-DKCMO exhibited greater stereoselectivity (>90% e.e., Table 3A). This gave the first indication that whereas the two DKCMOs lactonize the natural (*rac*)-monoterpenes camphor and diketocamphane with absolute divergent enantiocomplementarity, they can catalyse different patterns of biotransformation with other abiotic racemic ketone substrates. In order to explore this unexpected outcome more fully, both highly purified ketolactonases were challenged with a series of (*rac*)-*exo*-5-oxyfunctionalised bicyclo[2.2.1]heptan-2-ones [40], again producing a number of interesting outcomes (Table 3A). The nature of the substituting group determined whether either 2,5- or 3,6-DKCMO lactonized the substrate on a mutually exclusive basis, in each case generating a single optically pure bridgehead lactone in the same 1*S*,5*S*,6*R* enantiomeric series.

That idiosynchratic outcome was itself capitalised on to use the optically pure 1*S*,5*S*,6*R* bridgehead lactone generated from (*rac*)-exo-5-acetoxybicyclo[2.2.1]heptan-2-one by 2,5-DKCMO as a synthon for the chemical synthesis of the powerful insect antifeedant azadirachtin [92].

Both ketolactonases have been tested with a range of other abiotic alicyclic ketones, with the potential of generating synthons of value for chemoenzymatic synthesis as a principal driver. Given the proclivity of other BVMOs to generate homochiral products possessing three or four contiguous asymmetric carbon centres from various 2- and/or 3-substituted (*rac*)-bicyclo[3.2.0]heptan-6-ones [5,37,91,93,94], there are surprisingly few reports of equivalent studies undertaken with the ketolactonases. The characterised products (Table 3B; [4,38,91]) confirm that, with the exception of the most substituted substrate 7,7-dimethylbicyclo[3.2.0]hept-2-en-6-one, both 2,5- and 3,6-DKCMO generate the same (1*R*,5*S*)-2-oxa- and (1*S*,5*R*)-3-oxa-lactones by enantiodivergent biotransformations. With all the tested substrates, the e.e.% of both the 2-oxa- and 3-oxa-lactones produced by 2,5-DKCMO were significantly higher than the equivalent 3,6-DKCMO-generated products. These data have been interpreted as confirming that the active site of the 3,6-diketocamphane 1,6-monooxygenase is larger than that of the enantiocomplementary ketolactonase (*vide supra*, Figure 10). Although racemic alkyl-substituted cyclopentanones and cyclohexanones have been tested extensively with other BVMOs as a potential means for generating chiral lactones of value either as fragrance or flavouring agents or as synthons for chemoenzymatic synthesis [95,96], the only equivalent reported study on the ketolactonases was that conducted by Iwaki et al. [4]. While both recombinantly expressed isoenzymic 2,5-DKCMOs yielded the same 2-oxa-lactone products with similar yields and purity values, the only tested monocyclic ketone accepted by 3,6-DKCMO was 2-methylcyclohexanone, which was converted in low yield to the same (*S*)-series caprolactone product formed by both the enantiocomplementary isoenzymes. Again, despite the demonstrated ability of other BVMOs to desymmetrize prochiral 3-substituted cyclobutanones thereby generating chiral lactone products of value as synthons for the chemoenzymatic synthesis of lignans with anti-leukemic activities [97], only scant study has been made of equivalent biotransformations by highly purified ketolactonase preparations. Iwaki et al. [4] reported that 3,6-DKCMO was inactive with four such representative ketones, whereas both recombinantly expressed 2,5-DKCMO isoenzymes yielded corresponding lactone products, albeit with significantly different yields and degrees of purity.

More indirectly, the successful outcomes from the various initiatives to exploit the DKCMOs to catalyse biotechnologically valuable ketone biotransformations contributed significant impetus to exploring the equivalent potential of OTEMO [5], the camphor-induced Type I BVMO from *P. putida* ATCC 17543, and other Type I BVMOs [98,99,100,101,102] which pose different challenges as all these enzymes are single-component FAD-dependent flavoprotein monooxygenases [10,15].

An interesting study not focussed on generating synthons of value for chemoenzymatic synthesis but rather specifically designed to probe functional aspects of the active sites of the ketolactonases was conducted by challenging the enzymes with the specifically designed tricyclic ketone (*rac*)-tricyclo[4.2.1.0]nonan-2-one [103]. As previously recorded with a number of both bicyclo[2.2.1]- and bicyclo[3.2.0]ketones (Table 3A), highly purified preparations of the two ketolactonases proved to be stereochemically congruent by generating exclusively the same (−)-2-oxa-tricyclo[5.2.1.0]decan-3-one lactone in both very high purity (>98% e.e.) and yield (Figure 14). This exquisite outcome was interpreted to indicate that steric constraints imposed by the active sites of each DKCMO have little or no effect on stereoselectivity, but rather diastereofacial selectivity in the addition of the active oxygen co-substrate to promote ring expansion of the tricyclic ketone was paramount. A related study of cyclohexanone monooxygenase from *Acinetobacter calcoaceticus* NCIMB 9871 based on a range of other relevant substrates reached a similar conclusion for this single-component FAD-dependent Type I BVMO [104].

Finally, exploration of the chemoselectivity of ketolactonases in undertaking electrophilic oxygenations has received only limited attention. While the ability of the single-component FAD-dependent Type I BVMO cyclohexanone monooxygenase from *Acinetobacter calcoaceticus* NCIMB 9871 to biooxygenate various heteroatoms (S, Se, N, P, B, I) has been recognised for over 30 years [105], and the ability of microbial two-component FAD-dependent flavoprotein monooxygenases to catalyse enzymic sulfur-oxygenation documented more recently [106,107,108,109], the only relevant studies of the two-component FMN-dependent ketolactonases have been on the biooxidation of a range of organosulfides to equivalent chiral sulfoxides [39,110]. Significantly, when tested with twenty-three different organosulfides of various different types (Table 3D), no equivalent sulfone production, indicative of two successive biooxidation reactions, was detected with highly purified preparations of either enantiocomplementary ketolactonase activity. For the majority of the sulphides [110], both enantiocomplementary ketolactonases yielded chiral sulfoxides in the same enantiomeric series, albeit 2,5-DKCMO consistently generated products of higher chiral purity. In this respect these electrophilic oxygenation outcomes mirror those recorded from the various nucleophilic oxygenations catalysed by the same activities. The outcomes from this comprehensive study were used to produce the cubic-space models which suggest that 3,6-diketocamphane 1,6-monooxygenase has a larger active site than that of the enantiocomplementary ketolactonase (*vide supra*, Figure 10). This may in turn help to explain why this enzyme exhibits lower aspects of selectivity in undertaking not only electrophilic (sulfides to sulfoxides) but also nucleophilic (ketones to lactones) oxygenations (*vide supra*, Table 2). Interestingly, the luciferases from *Photobacterium phosphoreum* NCIMB 644 and *Vibrio fischeri* ATCC 7744, the only other two-component FMN-dependent monooxygenases with which comparable studies have been reported, both failed to biooxygenate a more limited range of dialkyl- and alkylarylsulfides [88].

There are well established roles both for enantiomerically pure sulfoxides as efficient chiral auxiliaries in the asymmetric synthesis of valuable target molecules [111,112], and for corresponding biotransformations catalysed by BVMOs other than the DKCMOs to generate these useful chiral moieties [113], which in some cases has been used as a means to increase the efficacy of a number of antitubercular thiocarbamide prodrugs [114,115]. An equivalent sulfoxidation of significant commercial value was developed recently by Codexis [116] using a genetically engineered BVMO for the large-scale production of the powerful proton-pump inhibitor esomeprazole (Nexium). Despite these highly relevant precedents, to date there have been no equivalent applications reported for sulfoxidations catalysed specifically by either of the enantiocomplementary ketolactonases. While a pilot study [117] did record the ability of both 2,5- and 3,6-DKCMO to generate the anticarcinogenic compound (−)-1-isothiocyanato-(4*R*)-(methylsulfinyl)butane (sulforaphane, CH_3_-SO-(CH_2_)_4_-NCS, [118,119]) to differing chiral purities (45% e.e. [24% conversion] and 51% e.e. [43% conversion] respectively) from erucin, the corresponding isothiocyanate precursor isolated from *Eruca sativa* seedmeal, this potentially valuable biotransformation currently remains unexploited.

## Figures and Tables

**Figure 1 microorganisms-07-00001-f001:**
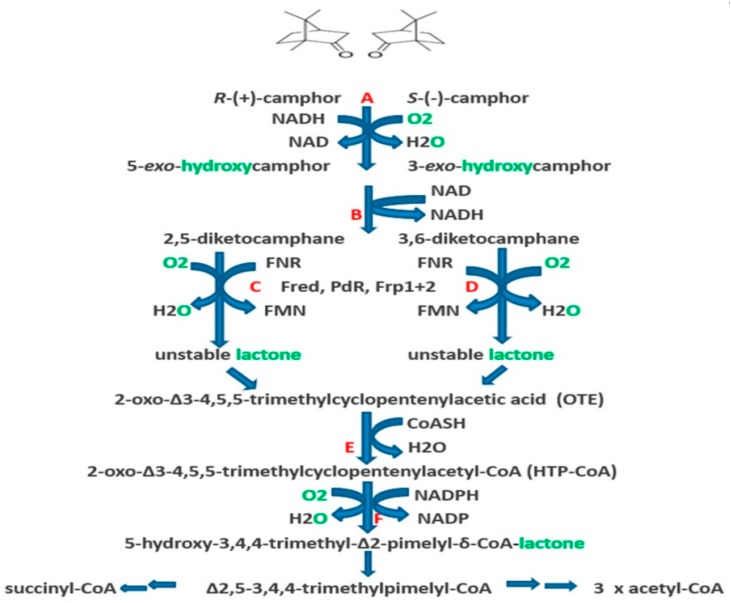
Pathway of (+)- and (−)-camphor degradation in *P. putida* ATCC 17453. A = cytochrome P450 monooxygenase (*camCAB*): B = *exo*-hydroxycamphor dehydrogenase (*camD*): C = 2,5-diketocamphane 1,2-monooxygenase (*camE*_25-1_ + *camE_25-2_*): D = 3,6-diketocamphane 1,6-monooxygenase (*camE*_36_): E = 2-oxo-∆^3^-4,5,5-trimethylcyclopentenylacetyl-CoA synthetase (*camF*_1_ + *F*_2_); F = 2-oxo-∆^3^-4,5,5-trimethylcyclopentenylacetyl-CoA monooxygenase (*camG*): FNR = reduced flavin mononucleotide: Fred = 36 kDa chromosome-coded flavin reductase: PdR = putidaredoxin reductase subunit of cytochrome P45O monooxygenase (*camA*): Frp 1 + 2 = chromosome-coded ferric reductases: diatomic oxygen molecules participating in the four monooxygenase-catalysed steps is shown in green, as in each case are the fates of each component oxygen atom.

**Figure 2 microorganisms-07-00001-f002:**
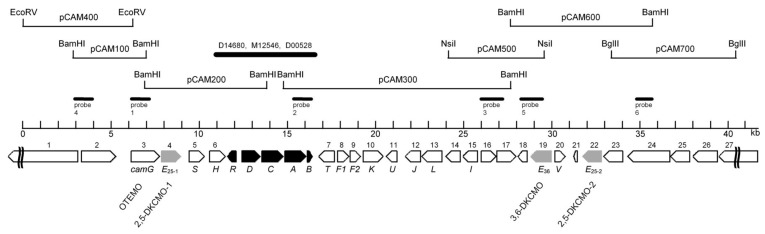
Localization of additional genes and predicted open reading frames (ORFs) flanking the established initial genes of the camphor *camDCAB* operon and its repressor, *camR*, on an ~40.5-kb sequenced region of the CAM plasmid of *P. putida* ATCC 17453. The predicted ORFs or genes are numbered from 1 to 27, except for the established *camRDCAB* genes, which are shaded in black. The orientation of the arrows indicates the direction of gene transcription. The candidate genes of this study (*camE*_25-1_, *camE*_25-2_, and *camE*_36_) representing the three diketocamphane monooxygenase (DKCMO) isozymes are highlighted in gray. The previously established OTEMO-encoding gene has been designated *camG* in accordance with the respective catabolic steps. *camS*, -*T*, -*U*, and -*V* are potential transcriptional regulators; *camV* is a close homolog of *camR*. The black solid line represents the previously sequenced region, with the indicated GenBank accession numbers. Figure 2 reproduced from Iwaki et al. [4], with the permission from American Society for Microbiology: licence number 4487631509592.

**Figure 3 microorganisms-07-00001-f003:**
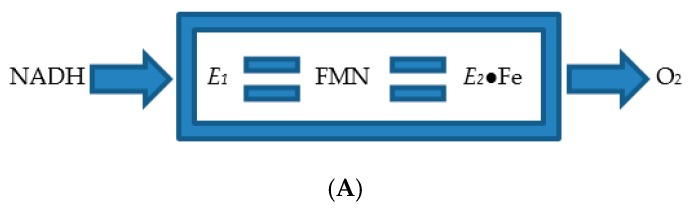
(**A**) Proposed [25] functional complex between *E*_1_ (NADH oxidase) and *E*_2_ (2,5-diketocamphane 1,2-monooxygenase); (**B**) Proposed [25] reaction sequence for the combined action of *E*_1_ and *E*_2_.

**Figure 4 microorganisms-07-00001-f004:**
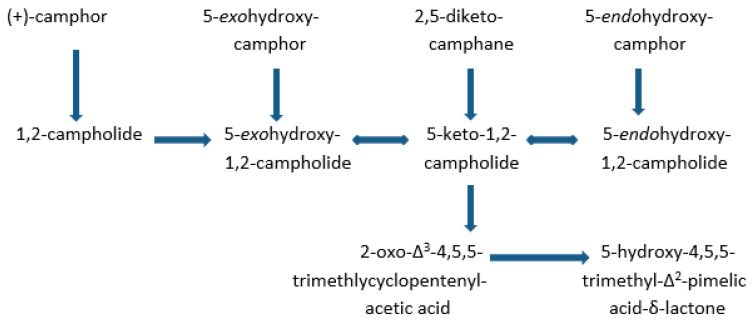
Speculative ‘metabolic grid’ proposed for the early metabolism of (+)-camphor [24,25,27] based on the characterised lactonizing reactions catalysed by partially purified *E*_2_ (2,5-diketocamphane 1,2-monooxygnase).

**Figure 5 microorganisms-07-00001-f005:**
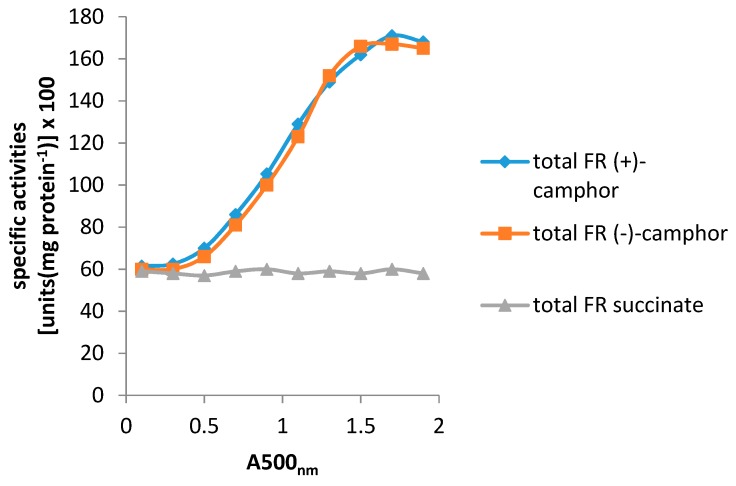
Specific activity of total flavin reductase (FR) throughout the growth of *P. putida* ATCC 17543 on either (+)-camphor, (−)-camphor or succinate as sole carbon source.

**Figure 6 microorganisms-07-00001-f006:**
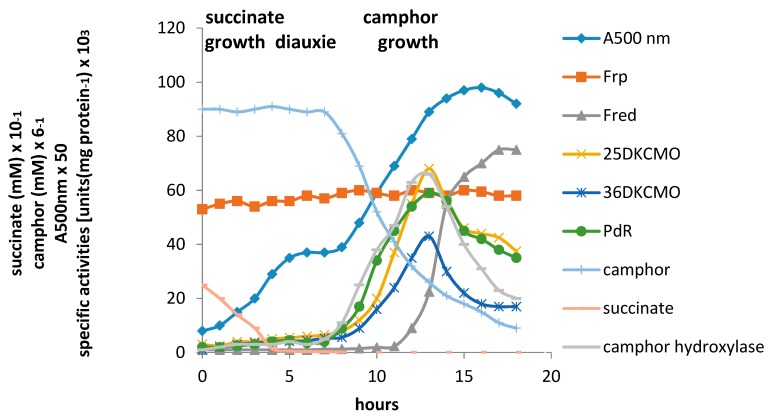
Changes in the optical density (A_500_nm), succinate (mM), (*rac*)-camphor (mM), and the specific activity of key enzymes of (*rac*)-camphor degradation during diauxic growth of *P. putida* ATCC 17543 on succinate plus (*rac*)-camphor-based defined medium.

**Figure 7 microorganisms-07-00001-f007:**
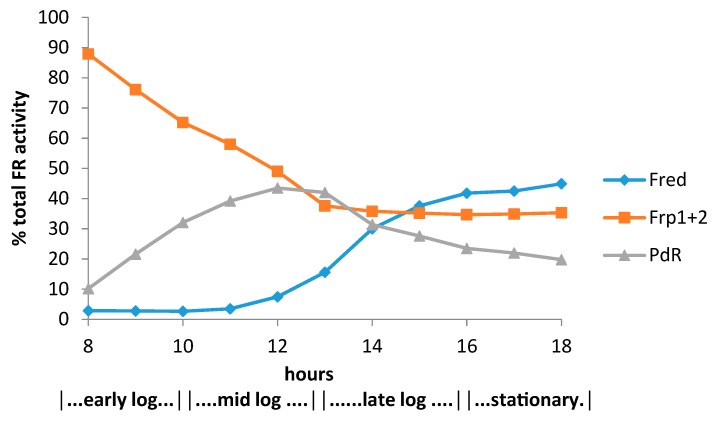
Relative contribution of the different assayed FNR-generating enzymes to the total flavin reductase activity titre throughout the various phases of (*rac*)-camphor-dependent growth of *P. putida* ATCC 17543 [18].

**Figure 8 microorganisms-07-00001-f008:**
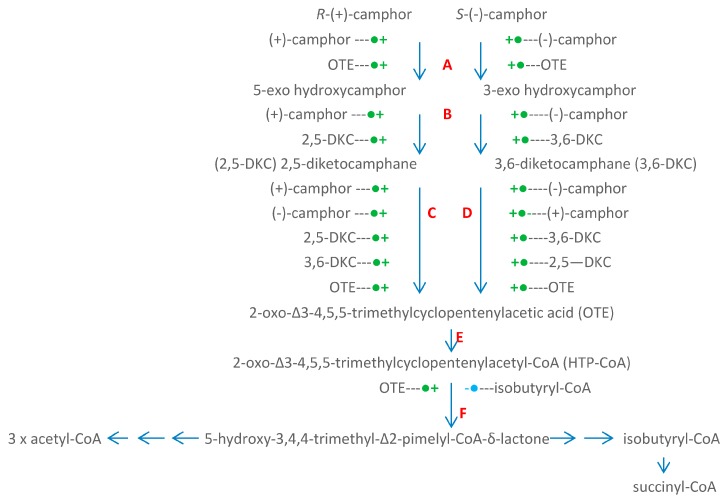
Transcriptional controls of the pathway of (+)- and (−)-camphor degradation in *P. putida* ATCC 17453 ---●**+** = induction: **-●**--- = repression: A = cytochromeP450 monooxygenase (*camCAB*): B = *exo*-hydroxycamphor dehydrogenase (*camD*): C = 2,5-diketocamphane 1,2-monooxygenase (*cam*_25-1_ + *cam*_25-2_): D = 3,6-diketocamphane 1,6-monooxygenase (*camE*_36_): E = 2-oxo-∆^3^-4,5,5-trimethylcyclopentenylacetyl-CoA synthetase (*camF*1 + *F*2): F = 2-oxo-∆^3^-4,5,5-trimethylcyclopentenylacetyl-CoA monooxygenase (*camG*) [59].

**Figure 9 microorganisms-07-00001-f009:**
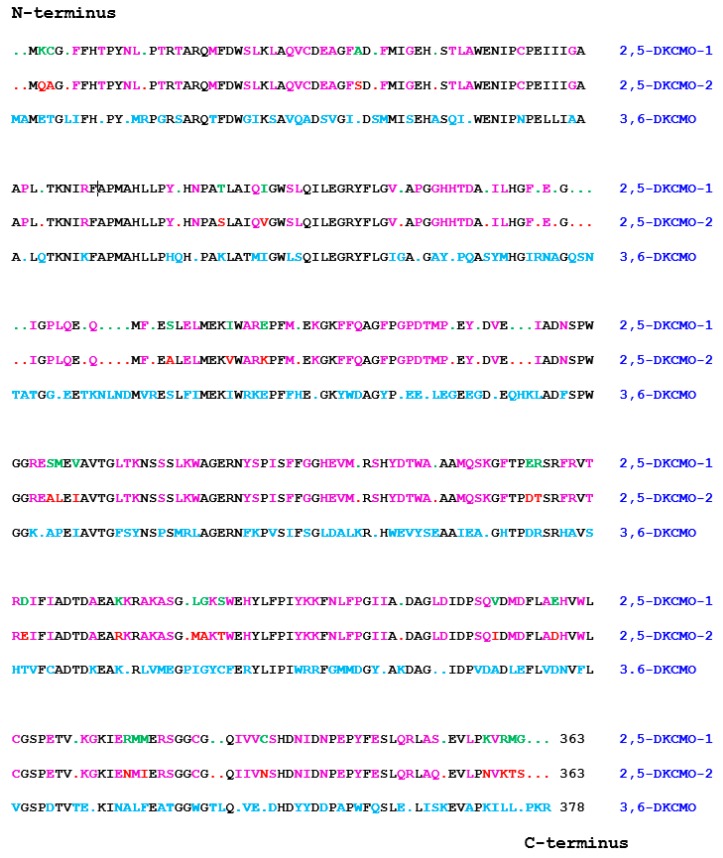
Protein sequence alignment of the oxygenating subunits of 2,5-DKCMO-1 (*orf-4*, *camE*_25-1_), 2,5-DKCMO-2 (*orf-22*, *camE*_25-2_) and 3,6-DKCMO (*orf-19*, *camE*_36_). Amino acid residues: common to 2,5-DKCMO-1, 2,5-DKCMO-2, and 3,6-DKCMO = black; common to 2,5-DKCMO-1 and 2,5-DKCMO-2 = purple; exclusive to 2,5-DKCMO-1 = green; exclusive to 2,5-DKCMO-2 = red; exclusive to 3,6-DKCMO = blue.

**Figure 10 microorganisms-07-00001-f010:**
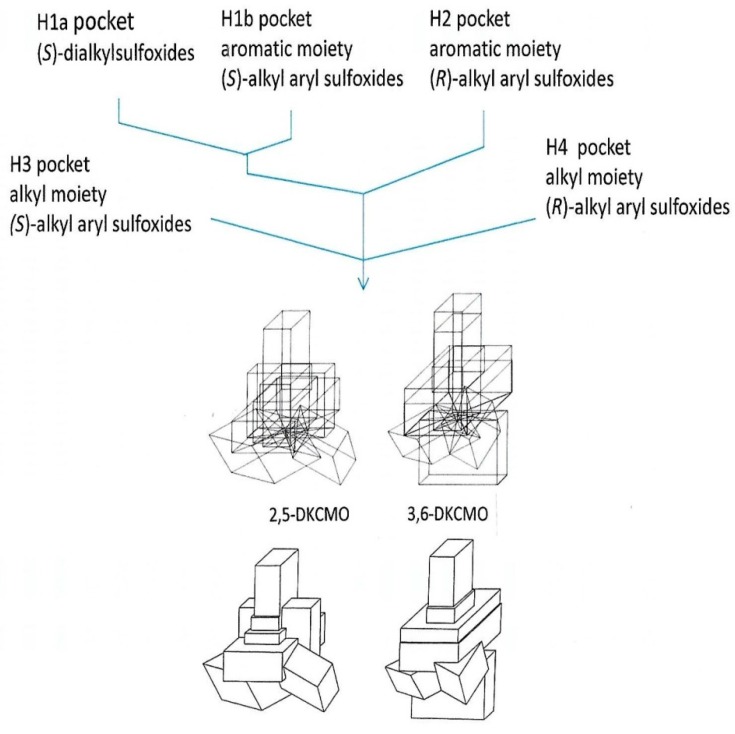
Rationale for superimposing the minimalised structures of the sulfoxides formed by 2,5-DKCMO and 3,6-DKCMO, and the resultant total dimensions (‘cubic space’) of the relevant enzyme active sites.

**Figure 11 microorganisms-07-00001-f011:**
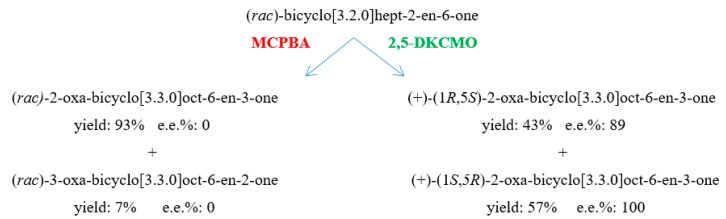
Contrasting outcomes of the lactonization of (*rac*)-bicyclo[3.2.0]hept-2-en-6-one by chemical oxidation (metachlorperbenzoic acid, MCPBA) and biotransformation (2,5-DKCMO); e.e.% = enantiomeric excess.

**Figure 12 microorganisms-07-00001-f012:**
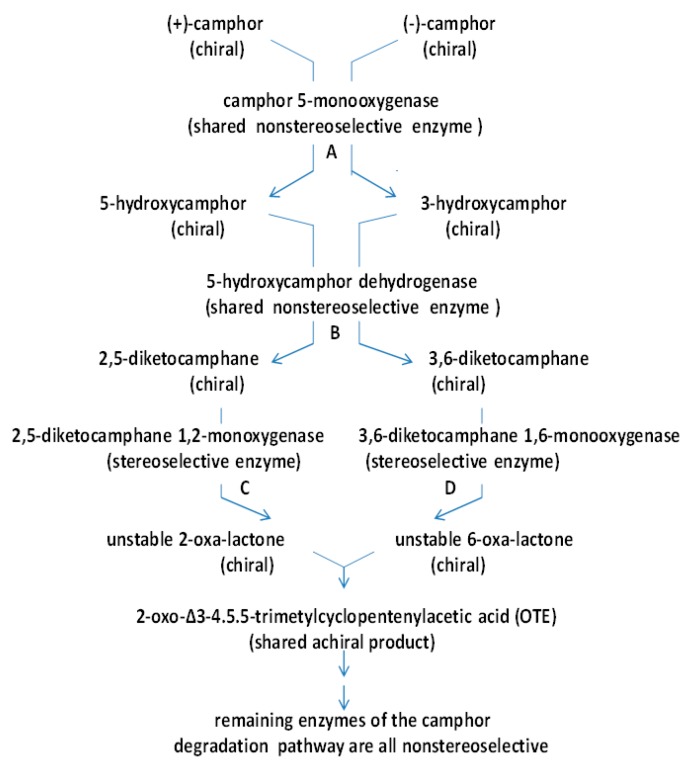
The relationships between nonselective and stereoselective enzymes that constitute the pathway for the degradation of (*rac*)-camphor to OTE in *P. putida* ATCC 17453, and the roles of relevant chiral and achiral molecules as substrates and transcriptional regulators. A = induction by (+)-camphor and (−)-camphor, plus product induction by OTE: B = induction by (+)-camphor and (−)-camphor, plus product induction by 2,5-DKC and 3,6-DKC: C = induction by (+)-camphor, (−)-camphor and 2,5-DKC, plus cross-induction by 3,6-DKC, and product induction by OTE: D = induction by (+)-camphor, (−)-camphor and 3,6-DKC, plus cross-induction by 2,5-DKC, and product induction by OTE: A + B + C + D = ‘from the top’ coordinate induction by (+)-camphor and (−)-camphor.

**Figure 13 microorganisms-07-00001-f013:**
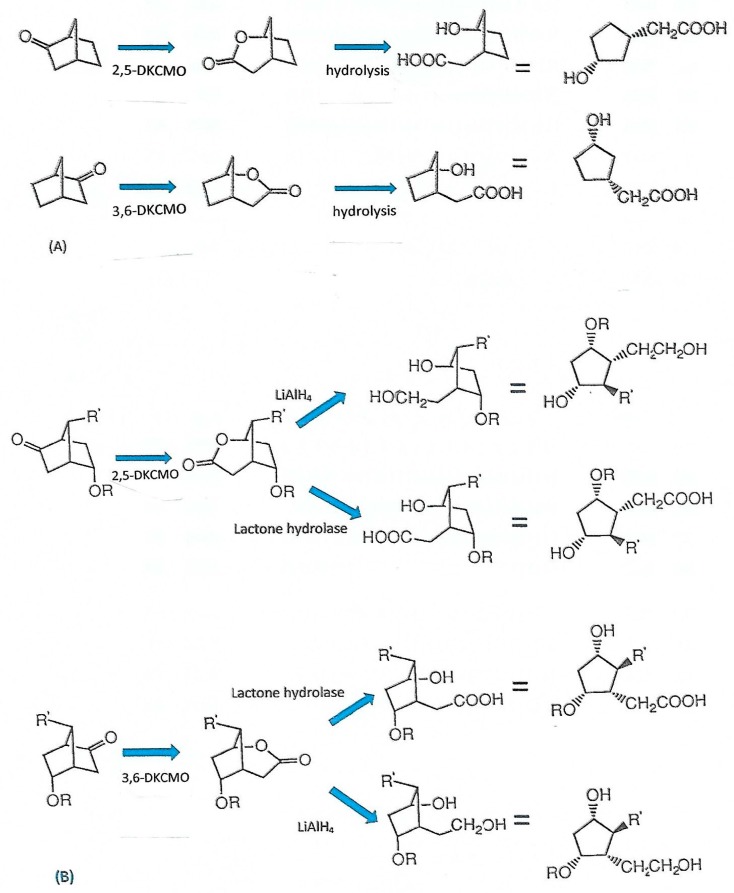
(**A**). Generation of enantiocomplementary substituted cyclopentanes with two chiral centres by deploying DKCMO-catalysed biotransformations of unsubstituted (*rac*)-bornan-2-one as a key step. (**B**). Generation of enantiocomplementary substituted cyclopentanes with four chiral centres by deploying DKCMO-catalysed biotransformations of (*rac*)-5′,7′-disubstituted bicyclo[2.2.1] ketones as a key step.

**Figure 14 microorganisms-07-00001-f014:**
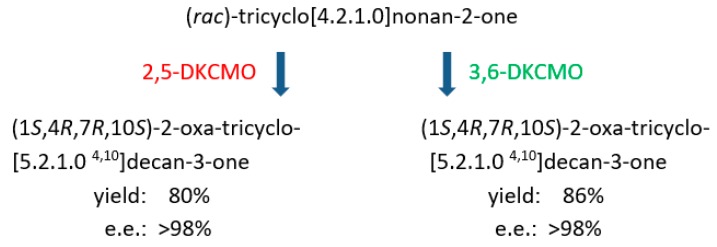
Biotransformation of the tricyclic ketone (*rac*)-tricyclo[4.2.1.0]nonan-2-one by highly purified preparations of native 2,5-DKCMO and 3,6-DKCMO; e.e. = enantiomeric excess.

**Table 1 microorganisms-07-00001-t001:** Cofactor content of purified ketolactonase *E*_2_ (2,5-diketocamphane 1,2-monooxygenase).

Cofactor	Moles/Mole Enzyme
Total iron	1.01
Heme iron	0.015
Flavin (FMN)	0.094
Molybdenum	<0.001
Copper	0.076

**Table 2 microorganisms-07-00001-t002:** Biotransformations of bicyclo[3.2.0] series ketones by highly purified preparations of 2,5-diketocamphane 1,2-monooxygenase and 3,6-diketocamphane 1,6-monooxygenase from camphor-grown *Pseudomonas putida* ATCC 17543.

Substrate	2,5-DKCMO-Generated Oxygenated Product(s)		3,6-DKCMO-Generated Oxygenated Product(s)	
7-*endo*-methylbicyclo[3.2.0]hept-2-en-6-one	(1*S*,5*R*)-3-oxa-lactone	(1*R*,5*S*)-2-oxa-lactone	(1S,5R)-3-oxa-lactone	(1*R*,5*S*)-2-oxa-lactone
	>95%, e.e.	>80%, e.e.	5%, e.e.	10%, e.e.
	100% conversion		100% conversion	
7,7-dimethylbicyclo[3.2.0]hept-2-en-6-one	(1*S*,5*R*)-3-oxa-lactone	(1*R*,5*S*)-2-oxa-lactone	racemic 3-oxa-lactone	no 2-oxa-lactone detected
	>80%, e.e.	>95%, e.e.		
	100% conversion		100% conversion	
methyl-para-tolyl sulfide	(*S*)-sulfoxide		(*S*)-sulfoxide	
	75%, e.e.		40%, e.e.	
	20% conversion		30% conversion	

e.e. = enantiomeric excess.

**Table 3 microorganisms-07-00001-t003:** Recorded outcomes of biotransformations of bicyclic ketones (**A**), other alicyclic ketones (**B**), aliphatic ketones (**C**), and organosulfides (**D**) by highly purified native or recombinantly expressed 2,5-DKCMO and 3,6-DKCMO.

Substrate	Enzyme			
	2,5-DKCMO-1 ^a^	2,5-DKCMO-2 ^b^	2,5-DKCMO-1 + 2 ^c^	3,6-DKCMO ^d^
(**A**)				
(+)-camphor product(s)	2-oxa-lactone	2-oxa-lactone	2-oxa-lactone	none
conversion %	100	100	100	0
e.e.%	>97	>97	>95	n.a.
(−)-camphor product(s)	none	none	none	6-oxa-lactone
conversion %	n.a.	n.a	n.a	100
ee%	n.a.	n.a	n.a	>97
2,5-diketocamphane product(s)	n.t	n.t	OTE(indirect)	none
conversion%			100	n.a.
e.e.%			n.a.	n.a.
3,6-diketocamphane product(s)	n.t	n.t	none	OTE(indirect)
conversion %			n.a.	100
e.e.%			n.a.	n.a.
(+)-fenchone product(s)	none	none	none	none
conversion %	n.a.	n.a	n.a	n.a
ee%	n.a.	n.a.	n.a.	n.a.
(−)-fenchone product(s)	none	none	none	none
conversion %	n.a	n.a	n.a	n.a
e.e.%	n.a.	n.a.	n.a	n.a.
(+)-nopinone product(s)	none	none	n, t	none
conversion %	n.a.	n.a		n.a
ee%	n.a.	n.a		n.a.
(*rac*)-bicyclo[2.2.1]heptan-2,5-dione product(s):	2-oxa-lactone	n.t	n.t.	2-oxa-lactone
conversion %	94			26
e.e.%	n.t			n.t.
6-oxo-cineole product(s)	n.t	n.t	6-oxa-lactone	n.t
conversion%			25	
e.e.%			n.t	
(*rac*)-bicyclo[3.2.0]hept-2-en-6-one product(s)	i. (+)-2-oxa-lactone (1*R*,5*S*)	i. (+)-2-oxa-lactone (1*R*,5*S*)	i. (+)-2-oxa-lactone (1*R*,5*S*)	i. (+)-2-oxa-lactone (1*R*,5*S*)
	ii. (+)-3-oxa-lactone (1*S*,5*R*)	ii. (+)-3-oxa-lactone (1*S*,5*R*)	ii. (+)-3-oxa-lactone (1*S*,5*R*)	ii. (+)-3-oxa-lactone (1*S*,5*R*)
conversion %	i. 55	i. 50	i. 43	i. 18
	ii. 35	ii.50	ii. 57	ii. 22
e.e.%	i. 77	i. 87	i. 89	i. 33
	ii. 99	ii. 97	ii. 100	ii. 82
(*rac*)- bicyclo[2.2.1]heptan-2-one(norcamphor) product(s)	i. (−)-2-oxa-lactone (1*R*,5*S*)	i. (−)-2-oxa-lactone (1*R*,5*S*)	(−)-2-oxa-lactone (1*R*,5*S*)	(−)-2-oxa-lactone (1*R*,5*S*)
	ii (−)-3-oxa-lactone (1*R*,5*S*)	ii (−)-3-oxa-lactone (1*R*,5*S*)		
conversion %	i. 48	i. 35	20	37
	ii. 16	ii. 25		
e.e.%	i. 40	i. 58	60	>90
	ii. 57	ii. 20		
(*rac*)-5-*exo*-hydroxybicyclo[2.2.1]heptan-2-one product(s)	n.t.	n.t.	none	(−)-2-oxa-lactone (1*R*,5*S*)
conversion %			n.a	33
ee			n.a.	>95
(*rac*)-5-*exo*-acetoxybicyclo[2.2.1]heptan-2-one product(s)	n.t.	n.t	(−)-2-oxa-lactone (1*R*,5*S*)	none
conversion %			35	n.a.
e.e.%			>95	n.a
(**B**)				
Cyclobutanone product(s)	none	n.t.	n.t.	2-oxa-lactone
conversion %	n.a.			13
e.e.%	n.a.			n.a.
Cyclopentanone product(s)	none	n.t	n.t	2-oxa-lactone
conversion %	n.a.			24
e.e.%	n.a.			n.a.
2-methylcyclopentanone product(s)	none	none	n.t.	none
conversion %	n.a.	n.a.		n.a.
e.e.%	n.a.	n.a.		n.a.
2-*n*-hexylcyclopentanone product(s)	(+)-2-oxa-lactone	(+)-2-oxa-lactone	n.t.	none
conversion %	6	3		
*E*.	12	5		
2-cyclopenten-1-one product(s)	n.t.	n.t.	n.t.	none
conversion %				n.a.
e.e.%				n.a.
3-methyl-2-cyclopenten-1-one product(s)	n.t.	n.t.	n.t.	none
conversion %				n.a.
e.e.%				n.a
2,3,4,5-tetramethyl-2-cyclopenten-1-one product(s)	n.t.	n.t.	n.t.	none
conversion %				n.a.
e.e.%				n.a.
Cyclohexanone product(s)	none	n.t.	n.t.	2-oxa-lactone
conversion %	n.a.			3
e.e.%	n.a.			n.a.
2-cyclohexen-1-one product(s)	n.t.	n.t.	n.t.	none
conversion %				n.a.
e.e.%				n.a.
2-methylcyclohexanone product(s)	(+)-2-oxa-lactone	(+)-2-oxa-lactone	n.t.	(+)-2-oxa-lactone
conversion %	8	4		2
*E*	3.2	3.1		3.2
2-ethylcyclohexanone product(s)	(+)-2-oxa-lactone	(+)-2-oxa-lactone	n.t.	none
conversion %	11	6		n.a.
*E*	43	22		n.a.
2-*n*-propylcyclohexanone product(s)	(+)-2-oxa-lactone	(+)-2-oxa-lactone	n.t.	none/trace
conversion %	18	13		n.a.
*E*	19	8.5		n.a.
2-phenylcyclohexanone product(s)	(−)-2-oxa-lactone	(−)-2-oxa-lactone	n.t.	(*rac*)-2-oxa-lactone
conversion %	11	6		2
*E*	43	22		n.a.
4-methylcyclohexanone product(s)	(−)-2-oxa-lactone	(−)-2-oxa-lactone	n.t.	trace
conversion %	8	5		n.a.
e.e.%	27	55		n.a.
4-ethylcyclohexanone product(s)	(−)-2-oxa-lactone	(−)-2-oxa-lactone	n.t.	(+)-2-oxa-lactone
conversion %	29	10		3
e.e.%	71	89		87
4-*n*-pentylcyclohexanone product(s)	(−)-2-oxa-lactone	trace	n.t.	trace
conversion %	2	n.a.		n.a
e.e.%	26	n.a.		n.a.
4-*tert-bu*tylcyclohexanone product(s)	(+)-2-oxa-lactone	trace	n.t.	none
conversion %	5	n.a.		n.a
e.e.%	61	n.a.		n.a.
3-methyl-2-cyclohexene-1-one product(s)	none	n.t.	n.t.	none
conversion %	n.a.			n.a.
e.e.%	n.a.			n.a.
3,5,5-trimethyl-2-cyclohexene-1-one product(s)	none	n.t.	n.t.	none
conversion %	n.a.			n.a.
e.e.%	n.a.			n.a
(**C**)				
Progesterone product(s)	none	n.t.	n.t.	n.t
conversion %	0			
e.e.%	n.a.			
2-decanone product(s)	none	n.t.	n.t.	n.c.
conversion %	0			11
e.e.%	n.a.			n.a.
Acetophenone product(s)	none	n.t.	n.t.	n.c.
conversion %	0			80
e.e.%	n.a.			n.a.
4-phenyl-2-butanone	none	n.t.	n.t.	n.c.
conversion %	0			48
e.e.%	n.a.			n.a.
(**D**)				
Methyl-*p*-tolyl sulphide product(s)	n.t.	n.t.	(*S*)-sulfoxide	(*S*)-sulfoxide
conversion %			12	57
e.e.%			62	32
*p*-methoxyphenyl methyl sulphide product(s)	n.t.	n.t.	(*S*)-sulfoxide	(*S*)-sulfoxide
conversion %			29	65
e.e.%			71	25
*p*-bromophenyl methyl sulphide product(s)	n.t.	n.t.	(*S*)-sulfoxide	(*S*)-sulfoxide
conversion %			29	65
e.e.%			71	25
*p*-chlorophenyl methyl sulphide product(s)	n.t.	n.t.	(*S*)-sulfoxide	(*S*)-sulfoxide
conversion %			5	15
e.e.%			45	27
*p*-fluorophenyl methyl sulphide product(s)	n.t.	n.t.	(*S*)-sulfoxide	(*R*)-sulfoxide
conversion %			9	72
e.e.%			39	30
*n*-pentyl methyl sulphide product(s)	n.t.	n.t.	(*S*)-sulfoxide	(*rac*)-sulfoxide
conversion %			28	90
e.e.%			32	n.a.
*n*-hexyl methyl sulphide product(s)	n.t.	n.t.	(*S*)-sulfoxide	(*S*)-sulfoxide
conversion %			19	50
e.e.%			37	19
*n*-heptyl methyl sulphide product(s)	n.t.	n.t.	(*S*)-sulfoxide	(*S*)-sulfoxide
conversion %			10	43
e.e.%			30	20
*n*-octyl methyl sulphide product(s)	n.t.	n.t.	(*S*)-sulfoxide	(*S*)-sulfoxide
conversion %			6	20
e.e.%			36	16
*n*-nonyl methyl sulphide product(s)	n.t.	n.t.	(*S*)-sulfoxide	(*R*)-sulfoxide
conversion %			2	9
e.e.%			16	8
*n*-decyl methyl sulphide product(s)	n.t.	n.t.	(*S*)-sulfoxide	(*R*)-sulfoxide
conversion %			1	9
e.e.%			5	9
*n*-octyl ethyl sulphide product(s)	n.t.	n.t.	(*S*)-sulfoxide	(*S*)-sulfoxide
conversion %			3	14
e.e.%			21	14
methyl phenyl sulphide product(s)	n.t.	n.t.	(*S*)-sulfoxide	(*S*)-sulfoxide
conversion %			9	53
e.e.%			35	9
ethyl phenyl sulphide product(s)	n.t.	n.t.	(*R*)-sulfoxide	(*rac*)-sulfoxide
conversion %			5	25
e.e.%			2	n.a.
*n*-propyl phenyl sulphide product(s)	n.t.	n.t.	(*R*)-sulfoxide	(*S*)-sulfoxide
conversion %			8	20
e.e.%			8	19
isopropyl phenyl sulphide product(s)	n.t.	n.t.	(*S*)-sulfoxide	(*R*)-sulfoxide
conversion %			11	21
e.e.%			20	24
*n*-butyl phenyl sulphide product(s)	n.t.	n.t.	(*R*)-sulfoxide	(*S*)-sulfoxide
conversion %			5	14
e.e.%			11	43
benzyl methyl sulphide product(s)	n.t.	n.t.	(*S*)-sulfoxide	(*R*)-sulfoxide
conversion %			29	85
e.e.%			30	4
benzyl ethyl sulphide product(s)	n.t.	n.t.	(*S*)-sulfoxide	(*R*)-sulfoxide
conversion %			20	41
e.e.%			2	9
2,3-dihydrobenzo-thiophene product(s)	n.t.	n.t.	(*S*)-sulfoxide	(*R*)-sulfoxide
conversion %			18	24
e.e.%			38	41
1-thiatetrahydro-naphthalene product(s)	n.t.	n.t.	(*S*)-sulfoxide	(*R*)-sulfoxide
conversion %			15	28
e.e.%			17	7

a = consolidated and averaged data from [4,35]; b = data from [4]; c = consolidated and averaged data from [6,17,37,38,39,40]; d = consolidated and averaged data from [4,6,17,36,37,38,39,40]; e.e.% = enantiomeric excess; *E* = enantiomeric ratio; n.a. = not applicable; n.c. = not characterised; n.t. = not tested.

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
