# Peer review of "Characterised Flavin-Dependent Two-Component Monooxygenases from the CAM Plasmid of Pseudomonas putida ATCC 17453 (NCIMB 10007): ketolactonases by Another Name"

_microorganisms, 2018, doi:10.3390/microorganisms7010001_

Round 1
Reviewer 1 Report
With great interest I have read this review about the two component type II BVMOs found in P. putida NCIMB 10007 as it is meticulously adding all the pieces of over 40 years of research together. Given all the different results published over time and all the missing and sometimes misleading information that has been published, I feel that it adds a lot of clarification to the nature of these enzymes and their activity and regulation.
I have a few comments listed below:
In the copy that I have reviewed, neither pages nor lines were numbered, which makes it hard for a reviewer to address the specific issues !
Page 2 (introduction) : there seems to be something missing "putative oxygen-" is followed by figure 1 but the text is not continued.
Figure 1: what does "FNR" stand for ? (releasing FMN)
Figure 2: where does it come from ? Iwaki et al. ?
Page 5: over table 1 correct text "to have a significant the inhibitory effect"
Page 5: below table 1, "studies have confirmed that none of these conclusions are incorrect" shouldn't it read "none...are correct" ??
Page 6: line 5: "containing a significant flavin content" what is significant ??
"using more powerful analytical techniques" please specify those techniques
Page 6: Figure 4 seems to be of low quality and can be hardly read
Page 6: under Figure 4: "and reported to be monomeric on the basis of a detection of a single N-terminal methionine by amino acid analysis" this does not make sense to me
Page 7: Line 4: SDS-PAGE as technique to investigate the quaternary structure ? I am not aware that SDS-PAGE could be used for quaternary protein structure assessment. It might be used in combination with other techniques, is this what the author meant to say ?
Page 16: "initial attempts to generate...crystals of 2,5-diketocamphane 1,2 MO were unsuccessful" Could this have been due to the two isoenzyme forms of 2,5-DKCMO (1 and 2) which would probably have co-eluted in conventional purification from P. putida ?
It should be emphasized in this review that references 31 and 32 (Kadow et al. 2011, and Kadow et al. 2012) cloned and expressed functional 2,5-DKCMO and 3,6-DKCMO, respectively, but that both studies failed to co-express the required FMN reductase which yielded in very low apparent activities (e.g. 0.9 mU/mg for +camphor ). Compared to later activities measured by Iwaki et al. (1U/mg) with a functional FMN-reductase (Fred) these "activities" by Kadow et al. were a magnitude of 1,000 times lower and thus substrate specificities described in those two paper should be disregarded as they seem to be not real activities but rather "noise". This would also explain the apparent specificity of 2,5-DKCMO towards (-)-camphor as reported in 31 that has been disputed in subsequent papers.
When referencing to et al. papers in the text "et al " (et alii) should be followed by a period "et al."
Author Response
I thank Reviewer 1 for their overall complementary appraisal and helpful specific comments. I have addressed these comments in the revised resubmitted manuscript as indicated below:
1. Page 2 (Introduction) …. something missing? The original formatting of Figure 1 caused a break in the relevant sentence which is in full ‘Additionally, there are two other putative oxygen-dependent enzymes coded for by orfs located within the 40.45-kb region of the CAM plasmid.’
2. Figure 1 ….. FNR? This is the reduced form of flavin mononucleotide (FMNH2) – the relevant abbreviation has been added to the legend of Figure 1.
3. Figure 2 … where does it come from? Iwaki et al. 2013 as now acknowledged in the resubmitted manuscript.
4. Page 5, over Table 1 …. incorrect text? Yes – amended manuscript now reads ‘to have a significant inhibitory effect’.
5. Page 6 lines 5 – 8 …. 2 questions regarding the flavin content of E2. These issues have been addressed by revising the relevant text to introduce more clarity.
6. Page 6 Figure 4 … low quality. This Figure, along with Figures 3A and 3B have now been revised to improve quality and clarity, and give greater conformity with Figure 1.
7. Page 6 plus Page 7 ….. analysis of E2. Use of the Edman reagent & degradation was the then used end-labelling method for N-terminal residue analysis of peptides/proteins. For reasons unknown, Trudgill et al. detected only 1 N-terminal residue (1 x methionine) per mole of protein (the method is notoriously inaccurate) – hence their conclusion that E2 was monomeric. SDS-PAGE - which Trudgill et al. never used to investigate E2 even after its widespread introduction in the late 1960s - would have enabled them to recognise the subunit structure of and hence homodimeric nature E2. The text has been modified to make these points clearer.
8. Page 16 .. initial attempts to generate crystals of 2,5-DKCMO. Reviewer 1’s suggestion is a valid one, and some relevant text has been included in the revised manuscript.
9. General comment with respect to the very low activity of the recombinant enzymes reported in the papers from the U of Greifswald (Kadow et al.). Reviewer 1 makes an important point which has been included in the revised manuscript (I trust the word “noise” is acceptable to the publishers!), along with new related information relevant to the two-component FMN-dependent monooxygenase luciferase from Photobacterium phosphoreum.
Reviewer 2 Report
Dear Author!
An intersting historical view on this enzyme
- In the introduction you refer to type 2 BVMOs and place a single reference; however, there are some more recent reviews available from the van Berkel group on flavoproteins in general and on two-component systems as well; this would add to your historical outline. … however, no need to be added just to mention;
- Page 2; the last sentence seems not fixed and here there is a formatting issue. “… Alternatively, there are two …” seems to be continued in legend of Figure 1 … here otherwise the last two sentences are not useful.
- Figure 1 need to be improved; so it is rather a scheme and of low quality.
- Figure 2; legend: the “P. putida” needs to be italic style; this should be corrected throughout the manuscript and also sometimes “spaces” and punctuations as not correct.
- Figure 3A and B need to be improved; this seems a bit cryptic and reaction patterns as these redox cascades can be improved.
- Figure 4 is of poor quality; not readable .. molecular structures would improve the quality as well.
- Page 9; E. coli; space missing
- Page 9; 18,466 kDa is not correct; punctuation
- Figure 9; the alignment need to be prepared with a proper tool; lines should have strain and enzyme designations etc, letters need to be aligned by a proper font etc.
- A large part is focused on sulfoxidation but this is only a side activity; there are other two component monooxygenases doing this similarly and those as well as type 1 BVMOs should be compared … they are also not so selective as the data summarized herein.
Best
Author Response
I thank Reviewer 2 for their overall complementary appraisal and helpful specific comments. I have addressed these comments in the revised resubmitted manuscript as indicated below:
1. Introduction ….. comments relevant to the classification of flavoproteins in general. Reviewer 2 makes a valid point, and the text has been revised to place the DKCMOs in current as well as historical context.
2. Page 2 … last sentence. This is a formatting issue related to Figure 1 and its legend - the original formatting of Figure 1 caused a break in the relevant sentence (‘Additionally, there are two other putative oxygen-dependent enzymes coded for by orfs located within the 40.45-kb region of the CAM plasmid.’). This can be resolved by the publishing journal.
3. Figures 1, 3A, 3B, and 4. These figures have been revised, improved, and reformatted into a broadly consistent style with the intention of reinforcing an element of continuity in the corresponding narrative of the developing history of the DKCMOs.
4. Figure 2 legend …. “P. putida”. This has been corrected to italic style (P. putida), and the related issues of relevant “spaces” (eg Page 9, E. coli), and punctuations corrected throughout the revised manuscript.
5. Page 9 …… details of Fred. This has been corrected to the format used in the original publication (Iwaki et al., 2013).
6. Figure 9. The information content and relevant text has been corrected with respect to 2,5-DKCMO-2. The amino acid residues are aligned, and each line has now been labelled with the relevant enzyme designation. The explanatory text has been reformatted to improve the discrimination between idiosynchratic residues and those common to two or more of the proteins.
7. Electrophilic biooxygenations. The scope of text has been broadened to acknowledge parallels between the FMN-dependent DKCMOs and other relevant two-component flavoproteins. A balance has to be struck to maintain focus on the FMN-dependent DKCMOs, but reference to other two-component FMN-dependent monooxygenases has now been included - a similar approach was adopted in a recent review of the two-component FAD-dependent monooxygenases [15].